# CONSTRAINT-BASED GRAPH NETWORK SIMULATOR

## ABSTRACT

In the rapidly advancing area of learned physical simulators, nearly all methods train a forward model that directly predicts future states from input states. However, many traditional simulation engines use a constraint-based approach instead of direct prediction. Here we present a framework for constraint-based learned simulation, where a scalar constraint function is implemented as a trainable function approximator, and future predictions are computed as the solutions to a constraint satisfaction problem. We implement our method using a graph neural network as the constraint function and gradient descent as the constraint solver. The architecture can be trained by standard backpropagation. We test the model on a variety of challenging physical domains, including simulated ropes, bouncing balls, colliding irregular shapes and splashing fluids. Our model achieves better or comparable performance to top learned simulators. A key advantage of our model is the ability to generalize to more solver iterations at test time to improve the simulation accuracy. We also show how hand-designed constraints can be added at test time to satisfy objectives which were not present in the training data, which is not possible with forward approaches. Our constraint-based framework is applicable to any setting in which forward learned simulators are used, and more generally demonstrates key ways that learned models can leverage popular methods in numerical methods.

## 1 INTRODUCTION

Consider a bowling ball colliding with a bowling pin. You might explain this event as involving a pair of forces being generated, one which causes the pin to move, and the other which causes the ball to career away with a different direction and speed. This kind of intuitive cause-and-effect approach is analogous to physical simulators that apply an explicit forward model to calculate a future state directly from the current one, such as when numerically integrating discretized equations of motion.

An alternative, but equally valid, way to explain the collision is in terms of constraint satisfaction: the ball and pin cannot occupy the same location at the same time, and their combined energies and momenta must be conserved, so the post-collision trajectories are the only way the future can unfold without violating these constraints. This constraint-based approach is analogous to physical simulators that use an implicit function to model a system of constraints over the current and future states, and which generate a prediction by searching for a future state that respects all constraints.

Both families of simulators—those based on explicit, forward functions versus those which define the dynamics implicitly, via constraints—are widely used in physics, engineering, and graphics. In principle they can model the same types of dynamics, however they differ in how their respective predictions are computed and in practice strike different trade-offs that determine why one or the other is preferred in different domains. For example, explicit methods are popular for large systems with (mostly) independent local effects whose space and time derivatives are relatively smooth, and their accuracy can often be increased by discretizing space and time more finely. Implicit approaches are often preferred for systems with strong interactions, such rigid and stiff dynamics, and more accurate solutions can often be found by using more sophisticated constraint solvers or by increasing the computational budget (e.g., solver iterations) allocated to searching for solutions. In machine learning (ML), there have been rapid advances recently in methods for learning to simulate complex dynamic processes, however almost all (e.g., Sanchez-Gonzalez et al. (2020); Pfaff et al. (2021)) have focused on explicit forward model approaches, with few exceptions (Yang et al., 2020).

Here we present a framework for learning to simulate complex dynamics via constraint satisfaction. Our "Constraint-based Graph Network Simulator" (C-GNS) defines a single scalar-valued constraint function that represents whether a future state satisfies the physical constraints, conditioned on the current and previous states. The constraint function is implemented as a Graph Neural Network (GNN) (Bronstein et al., 2017; Battaglia et al., 2018), which can model systems with rich compositional structure—multiple bodies, complex meshes, etc. To predict the next state via the constraint function's implicit representation of the dynamics, a gradient-based solver finds a proposed state which satisfies the constraints. We train it through the solver by backpropagation. We also introduce a hybrid approach that proposes and refines the future state using an explicit iterative predictor, rather than solving for learned constraints.

We tested the C-GNS on a variety of challenging physical simulation domains generated by several different simulation engines: simulated rope, bouncing balls, and bouncing irregular rigid shapes (MuJoCo, Todorov et al. (2012)) and splashing fluids (Flex, Macklin et al. (2014)). We found that the C-GNS's simulated rollouts were more accurate than a state-of-the-art Graph Net Simulator (GNS) (Sanchez-Gonzalez et al., 2020) with comparable number of parameters. At test time, the C-GNS could use additional solver iterations to improve its predictive accuracy, striking desired speed-accuracy trade-offs. It could also satisfy new, hand-designed constraints jointly alongside its learned constraints. Neither of these capabilities are possible in explicit forward-style approaches.

## 2 BACKGROUND AND RELATED WORK

Constraint solvers are central to many physics simulators. Most rigid-body and game engines use constraints to model joints, collision and contact (Baraff, 1994). They are used for limiting strain in realistic cloth simulation (Thomaszewski et al., 2009), and are a core component in Eulerian incompressible fluid solvers to solve for pressure (Chorin, 1967). Recently, position-based (Müller et al., 2007) and projective dynamics methods (Bouaziz et al., 2014) have become very popular for interactive simulation. These methods express dynamics purely as constraints, and can simulate a wide range of physical systems from rigids over soft-bodies to fluids (Macklin et al., 2014).

Machine learning methods for accelerating scientific simulation of complex systems, such as turbulence (Kochkov et al., 2021; Wang et al., 2020) and aerodynamics (Thuerey et al., 2020; Zhang et al., 2018), have grown rapidly in recent years. GNN-based learned simulators, in particular, are a very flexible approach which can model a wide range of systems, from articulated dynamics (Sanchez-Gonzalez et al., 2018) to particle-based physics (Mrowca et al., 2018; Li et al., 2019; Sanchez-Gonzalez et al., 2020) and mesh-based continuum systems (Pfaff et al., 2021; De Avila Belbute-Peres et al., 2020), and generalize well to unseen scenarios. Combining learning algorithms with principles from physics and numerical methods, such as auxiliary loss terms and rich inductive biases, can improve sample complexity, computational efficiency, and generalization (Wu et al., 2018; Karniadakis et al., 2021; Chen et al., 2018; Rubanova et al., 2019). Imposing Hamiltonian (Greydanus et al., 2019; Sanchez-Gonzalez et al., 2019; Chen et al., 2019) and Lagrangian (Lutter et al., 2019; Cranmer et al., 2020; Finzi et al., 2020) mechanics in learned simulators offers unique speed/accuracy tradeoffs and can preserve symmetries more effectively.

Recent methods have been proposed for learning constraint functions and solving them in a model's forward pass (Duvenaud et al. (2020)'s "Deep Implicit Layers" tutorial is an excellent hands-on survey). Such models can play games (Amos & Kolter, 2017; Wang et al., 2019), optimize power flow (Donti et al., 2021), support robotic planning (Loula et al., 2020), and perform combinatorial optimization (Bartunov et al., 2020). Solvers such as gradient descent and Newton's method are differentiable, and support training by backpropagation, but this can be computationally expensive, so approaches such as Deep Equilibrium Models (DEM) (Bai et al., 2019; 2020) use implicit differentiation to compute gradients only at the solution point.

Despite the popularity of constraint-based traditional simulators, only a single simulator which uses learned constraints has been reported (Yang et al., 2020). Their "Neural Projections" method, based on Goldenthal et al. (2007), iteratively proposes a future state with an Euler step, then projects the proposal onto a learned constraint manifold, implemented as a multilayer perceptron (MLP). Crucially, their constraint function only measures how much an individual state violates the learned constraints, and thus is not an implicit representation of the dynamics. It is suitable for quasi-static regimes, but not scenarios such as the elastic collisions in the bowling ball example described above.

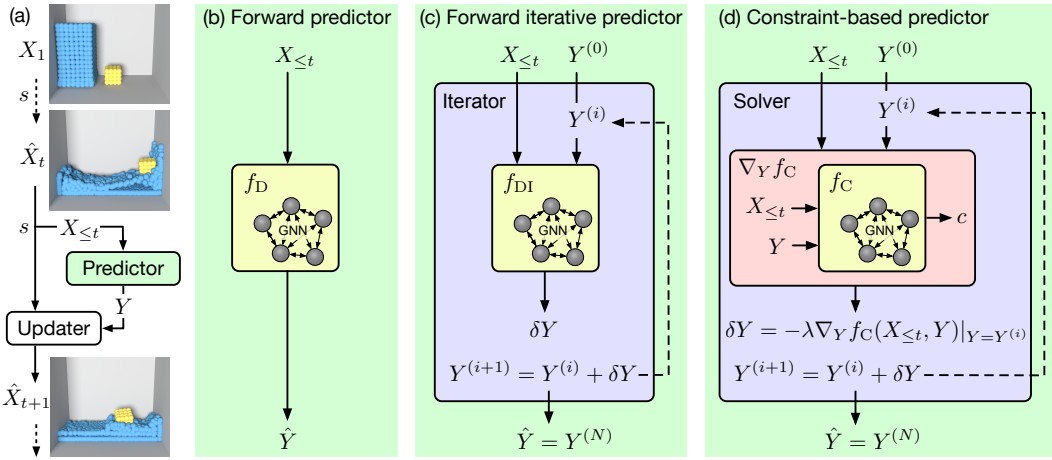

Figure 1: **Learned simulator schematics.** **(a)** A simulator, $s$, maps $X_{\leq t}$ to a future state $\hat{X}_{t+1}$, using a PREDICTOR that returns $\hat{Y}$, which represents information about the system's temporal evolution, and an UPDATER which uses $\hat{Y}$ to update $X_t$ to $\hat{X}_{t+1}$. **(b)** Forward GNN simulator. The PREDICTOR maps $X_{\leq t}$ directly to $\hat{Y}$ using $f_D$. **(c)** Iterative GNN simulator. The iterator refines $\hat{Y}$ by repeatedly applying $f_{DI}$. **(d)** Constraint-based Graph Network simulator (C-GNS). The PREDICTOR iteratively solves for a $\hat{Y}$ that satisfies a constraint function, $f_C$, using $\nabla_Y f_C$.

## 3 MODEL FRAMEWORK

**Simulation basics** A physical trajectory, measured at discrete time intervals, is a sequence of states, $(X_1, \ldots, X_T)$, where $X_t$ represents properties such as the positions, velocities, masses, etc, of elements of the system. A physical simulator, $s$, is a function that maps current and/or previous state(s), which we term the *context*, $X_{\leq t}$, to a predicted future state, $\hat{X}_{t+1} = s(X_{\leq t})$ (see Figure 1a)[1]. A simulated physical trajectory, termed a *rollout*, $(X_t, \hat{X}_{t+1}, \hat{X}_{t+2}, \ldots)$, can be generated by repeatedly applying $s$ to its own predicted state, $\hat{X}_{t+1} = s(\hat{X}_{\leq t})$.

Simulators are often comprised of a PREDICTOR mechanism which maps the context $X_{\leq t}$ to an update value $\hat{Y}$, that represents information about the system's temporal evolution at the current time. Then $\hat{Y}$ is used by an UPDATER mechanism to update the current state to the next state: $\hat{X}_{t+1} = \text{UPDATER}(X_{\leq t}, \hat{Y})$, e.g., updating current positions and velocities represented by $X_t$ with new velocities and accelerations represented by $\hat{Y}$, to predict the next state.

**Explicit simulators** Across science, engineering, and graphics, a popular class of simulators are defined *explicitly*: the state update $\hat{Y}$ is predicted directly from $X_{\leq t}$ using an explicit forward function, $\hat{Y} = f_D(X_{\leq t})$, as illustrated in Figure 1b. Among the rapidly growing family of learned simulators, the forward function $f_D$ is typically implemented using a neural network (Sanchez-Gonzalez et al., 2020; Pfaff et al., 2021).

**Constraint-based implicit simulators** Here we explore learned simulators based on *implicit* formulations of the dynamics. Rather than predicting the desired state directly, as in explicit formulations, our implicit simulator uses a differentiable constraint function, $c = f_C(X_{\leq t}, \hat{Y})$, where $c$ is a scalar that quantifies how well a proposed state update $\hat{Y}$ agrees with $X_{\leq t}$. A future prediction is generated by applying a solver, such as an optimization or zero-finding algorithm, to find a $\hat{Y}$ that satisfies the constraint function, and applying the UPDATER to update $X_t$ to $\hat{X}_{t+1}$. The $f_C$ can represent all the physical constraints in the system, including the time dynamics.

---

[1]Despite that physics is Markovian, we use $X_{\leq t}$ as input because our framework can also apply to dynamic processes which are non-Markovian. Providing previous states can also often be helpful when there are hidden properties of the system which are only identifiable over a sequence of observed states, and when a state does not represent velocity or momentum information.

As illustrated in Figure 1d, we formulate our constraint-solving procedure via an iterative method that starts with an initial proposal, $Y^{(0)}$. On the $i$-th iteration, the solver uses the gradient of $f_C$ w.r.t. $Y$ at the current proposal to compute a change to the proposal, $\delta Y = -\lambda \, \nabla_Y f_C(X_{\leq t}, Y)|_{Y=Y^{(i)}}$. This $\delta Y$ is then used to revise the proposal to, $Y^{(i+1)} = Y^{(i)} + \delta Y$. This process repeats for $N$ steps, and the final proposal value is treated as the PREDICTOR's output, $\hat{Y} = Y^{(N)}$.

Our constraint-based model's $f_C$ is defined as a trainable function approximator which is real-valued and lower bounded at zero, and uses gradient descent to find $\hat{Y}$ that minimizes it, where $\lambda$ is a fixed step size. This induces the semantics that the desired $\hat{Y} = \arg\min_Y f_C(X_{\leq t}, Y)$.

We also explore a second constraint-solving procedure, inspired by Yang et al. (2020)'s Neural Projections' use of "fast projection" Goldenthal et al. (2007). Specifically, $\lambda = -\frac{f_C(X_{\leq t}, Y^{(i)})}{\left\| \nabla_Y f_C(X_{\leq t}, Y)|_{Y=Y^{(i)}} \right\|^2}$. Unlike gradient descent, fast projection is a zero-finding algorithm, so in this case $f_C$ is not lower bounded. This induces the semantics that $f_C(X_{\leq t}, \hat{Y}) = 0$.

This general formulation of constraint-based learned simulation can be trained by backpropagating loss gradients through the solver loop[2]. The computational budget of the forward pass can be varied via the number of solver iterations $N$.

**Explicit iterative simulators** As a hybrid between forward and constraint-based simulators, we introduced a model which iteratively refines a proposed state update, like in the constraint-based approach described above, but using an explicit function to directly output a $\delta Y$ at each iteration, rather than solving a constraint function (see Figure 1c). See Section 4.3 for details.

## 4 EXPERIMENTS

### 4.1 EXPERIMENTAL TASK DOMAINS

We test our framework on a variety of physical environments, shown in Figure 2: ROPE, BOUNCING BALLS and BOUNCING RIGIDS, whose ground truth training and test data were generated by the MuJoCo physics simulator, as well as BOXBATH from Li et al. (2019). These environments demonstrate a diverse set of physical constraints: 'hard' constraints (preserving the shape of the rigid object and resolving collisions), and 'soft' constraints on fluid movement, handling gravity and preserving the momentum of the rope and bouncing balls. See the Supplementary Materials for details.

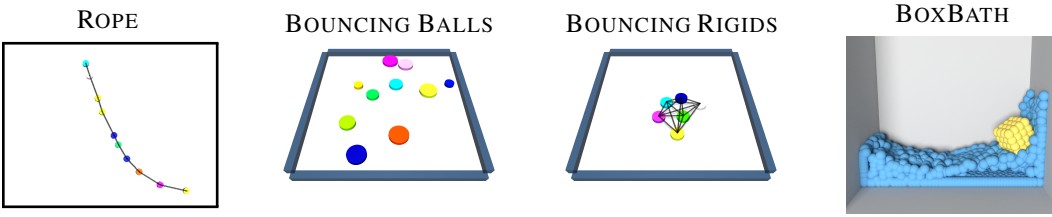

| ROPE | BOUNCING BALLS | BOUNCING RIGIDS | BOXBATH |

Figure 2: **Renderings of the physical environments.** Videos of the model rollours are available at: sites.google.com/view/constraint-based-simulator.

### 4.2 MODEL IMPLEMENTATIONS

**Representing the physical system** Our experimental domains are physical systems comprised of sets of interacting point-like elements, e.g., objects, particles, mesh vertices, etc. We represent the state as $X_t = (p_t^j)^{j=1...|X_t|}$, where $|X_t|$ is the number of elements, and $p_t^j$ is the $j$-th element's position at time $t$. There are also other static properties of the physical elements, e.g., masses, material types, etc., which we represent with $Z$ to keep it distinct from the dynamic state information represented by $X_t$. The input context is $X_{\leq t} = (Z, X_{t-3}, X_{t-2}, X_{t-1}, X_t)$.

---

[2]Implicit differentiation at the solution point should be applicable as well, and potentially offer computational benefits as mentioned in the Section 2, though we do not explore that here.

In our implementation, $\hat{Y}$, represents the predicted changes in position (i.e., the "average velocity" across the time step)[3], $\hat{y}^j = \Delta\hat{p}^j_{t+1} = \hat{p}^j_{t+1} - p^j_t$. The UPDATER then computes $\hat{X}_{t+1}$ using $\hat{p}^j_{t+1} = p^j_t + \Delta\hat{p}^j_{t+1}$, where $p^j_t$ is provided in the input $X_{\leq t}$.

**Constructing the input graph** Our implementations of the $f_D$, $f_{DI}$, and $f_C$ use GNNs as the function approximators, so we need to pack the context, $X_{\leq t}$, and (for the $f_{DI}$ and $f_C$) the proposed state update information, $Y^{(i)}$, into an input graph, $G_t = (V_t, E_t)$. The edges $E_t$ represent possible interactions among the elements, such as fully connected edges to represent collisions and rigid attachments in BOUNCING BALLS and BOUNCING RIGIDS, spring constraints in ROPE, and interactions among particles within a fixed connectivity radius in BOXBATH.

We enforced translation-invariance by construction, by never providing absolute positions as input to the models. Instead, the $j$-th input node's features are the static properties, and a sequence of the three most recent position changes (i.e. average velocities), $v^j_t = [z^j, \Delta p^j_{t-2}, \Delta p^j_{t-1}, \Delta p^j_t]$, where, $\Delta p^j_t = p^j_t - p^j_{t-1}$. For $f_{DI}$ and $f_C$, which also take the solver's current proposed $Y^{(i)}$, we also concatenate the proposed average velocity from the $i$-th solver iteration, $y^{j,(i)} - p^j_t$, as input. For the input edge feature for an edge that connects from node $j$ to $k$, we also provide the relative displacement vector between the nodes' positions, $e^{jk}_t = p^k_t - p^j_t$.

**GNN-based Encode-Process-Decode core** We implemented $f_D$, $f_{DI}$, and $f_C$ using Graph Networks (GN) (Battaglia et al., 2018), arranged in the Encode-Process-Decode architecture, similar to previous work on GN-based learned simulators (Sanchez-Gonzalez et al., 2018; 2020; Pfaff et al., 2021). The Encoder uses two MLPs to encode node and edge features into high-dimensional latent vectors. The Processor applies multiple GNs, with unshared weights, in sequence, with node and edge residual connections at each step. We do not use global updates for the GNs. The Decoder uses an MLP to produce an output for each node.

The $f_D$ directly returns $\hat{Y}$. The $f_{DI}$ returns a change to the proposed update $\delta Y$ for the current iteration. The $f_C$'s Decoder returns a scalar for each node to produce a constraint value per node $\{c^j | j = 1 \dots |V|\}$. These node-wise constraint values are averaged to compute a single scalar $c$ constraint for the entire system, $c = f_C(X_{\leq t}, \hat{Y}) = \frac{1}{|V|} \sum_{j=1}^{|V|} c^j$.

**Solving the constraint** For $f_{DI}$ and $f_C$ we initialize $Y^{(0)} = \Delta p^j_t$ to the most recent average velocity[4]. We used auto-differentiation in JAX to compute the gradient function, $\nabla_Y f_C$, and the step size $\lambda$ was specific to the model variant, as described below. During training we used $N = 5$ solver iterations.

### 4.3 MODEL VARIANTS

The key questions in this work are whether constraint-based learned simulators can compete with explicit, forward learned simulators, whether implementing the constraint function with GNNs is more effective than with MLPs, and how minima-based constraint functions solved by gradient descent compare to constraints defined as the zeros of a function which are solved by fast projection (Goldenthal et al., 2007). The following model variants allow us to answer these questions.

**Forward GNN** This is an explicit, forward GNN-based learned simulator based on the GNS models from Sanchez-Gonzalez et al. (2020); Pfaff et al. (2021). It directly predicts the state update $\hat{Y}$ from the past time points $X_{\leq t}$.

**C-GNS Gradient Descent (C-GNS-GD)** and **C-GNS-Fast Projections (C-GNS-FP)** These are our proposed constraint-based GNN models. For the C-GNS-GD, the scalar per-node output $c^j$ was squared, to force the overall $f_C$ to be non-negative, and a gradient descent solver with a fixed step size, $\lambda = 0.001$, was used to minimize it. For C-GNS-FP, the $\lambda$ was based on "fast projection" (Goldenthal et al., 2007; Yang et al., 2020), as described in Section 3. Supplementary Figure B.5(c-d) shows ablations.

---

[3]For BOXBATH we vary a number of modelling choices to best match those in Sanchez-Gonzalez et al. (2020). The major difference is that we set $\hat{Y}$ to be the average acceleration rather than average velocity. See Supplementary Materials for other differences.

[4]To ensure analogous information is provided downstream of $f_D$, the update rule also includes the previous average velocity: $\hat{p}^j_{t+1} = p^j_t + \Delta p^j_t + \hat{y}^j$

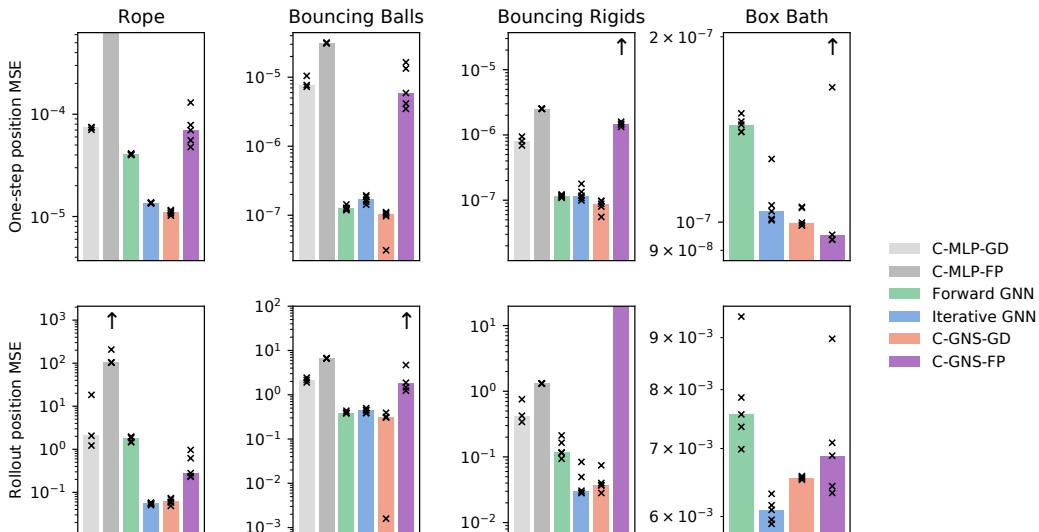

Figure 3: **Test MSE on the node positions predictions across different models.** Top row: 1-step position MSE. Bottom row: full 160-step rollout MSE. The bar height represents the median MSEs over random seeds. The black cross marks show the MSE metric for each random seed. The black arrows indicate if MSE metric for a random seeds exceeds the upper y limit of the figure.

**Iterative GNN** We implemented a hybrid between the Forward GNN and C-GNS, as shown in Figure 1c. It was identical to the C-GNS models, except its $f_{DI}$ directly predicted proposed state updates as in $f_D$, rather than being computed via the gradients as was done with $f_C$.

**ConstraintMLP Gradient Descent (ConstraintMLP-GD)** and **ConstraintMLP-Fast Projections (ConstraintMLP-FP)** These were MLP-based constraint models, which, rather than using GNNs to implement $f_C$, instead concatenated the embeddings of all the input nodes into a single vector and passed them to an MLP implementation of $f_C$. By default, these models cannot handle variable-length inputs, so we padded smaller states with zeros up to the maximum state size. The ConstraintMLP-FP was the MLP analog to our C-GNS-FP, and was similar to Neural Projections (Yang et al., 2020). The ConstraintMLP-GD used gradient descent, and was the MLP analog to our C-GNS-GD. We omit the results for the ConstraintMLP models on BOXBATH (1024 nodes), as MLPs do not generally work well on physical systems with more than a few particles (Battaglia et al., 2016; Sanchez-Gonzalez et al., 2018).

## 4.4 TRAINING AND EVALUATION

We trained the models to make next-step predictions, by computing the $L_2$ loss between the predicted $\hat{X}_{t+1}$ and the corresponding ground truth $X_{t+1}$, averaged over nodes. All model weights and biases were trained using standard backpropagation with the Adam optimizer.

At test time, we compute 1-step metrics by evaluating the 1-step errors along each point of the ground truth trajectory. We also evaluate rollout errors by iteratively applying the learned model starting from an initial state, over 160 rollout steps, and computing the error between the predicted and ground truth trajectories.

## 5 RESULTS

**Predictive accuracy**[5] Our experimental results show that our C-GNS-GD's performance was generally better than the other model variants. Figure 3 compares the different models on 1-step and rollout position MSE (see Supplementary Table B.1 for numerical results). For each dataset, we

---

[5]Videos of the model rollouts are available at sites.google.com/view/constraint-based-simulator

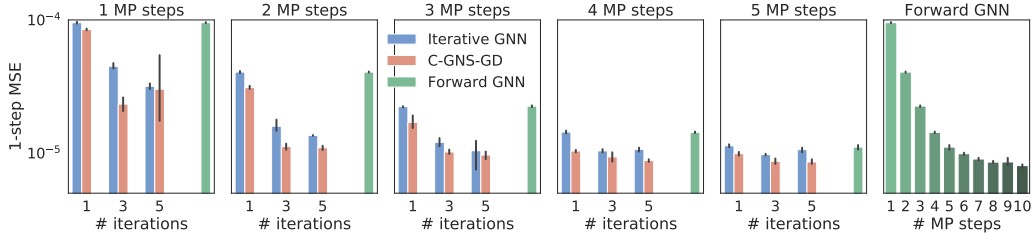

Figure 4: **Test 1-step MSE error on ROPE as a function of message-passing (MP) steps and solver iterations.** The left five subplots shows performance for different numbers of message-passing steps. The blue and red bars represent the Iterative GNN and C-GNS-GD, respectively, where the bar height is 1-step MSE. The green bars show the Forward GNN (it does not use solver iterations). The rightmost subplot shows the Forward GNN with 1 to 10 MP steps.

used the same number of message-passing steps (MP) for all GN-based models. We used 2 MPs for the ROPE dataset, and 1 MP for all other tasks.

The C-GNS-GD has lower 1-step MSE between the ground truth and predicted positions than other models across all datasets. Qualitatively, we observed that for Forward GNN with a single message-passing step, the box in BOXBATH "melts" over time, as the forward model cannot preserve its rigid shape (see Videos). The comparable C-GNS-GD, by contrast, maintains the rigidity more effectively. These quantitative results suggest that constraint-based learned simulators are competitive alternative to explicit, forward learned simulators. We generally found that the Iterative GNN was fairly competitive with the C-GNS-GD in overall performance and better than the Forward GNN.

We also found that the C-GNS-FP was generally less stable across seeds, and not as accurate as the C-GNS-GD. The same conclusion holds for ConstraintMLP-FP versus ConstraintMLP-GD. We speculate that the fast projection algorithm may make training challenging because the step size $\lambda$ is proportional to $f_C$, which may cause poor zero-finding early in training when the $f_C$ is not yet informative. Additionally, we find that C-GNS-FP algorithm becomes unstable in the areas with shallow constraint gradients, perhaps because its $\lambda$ depends on the inverse of the gradient's norm.

We explored how varying the message-passing steps and solver iterations ($N$) influenced the relative performance among the models in our ROPE dataset. Figure 4 shows that the C-GNS-GD generally required fewer parameters and message-passing steps to achieve comparable 1-step MSE to the other models. Supplementary Figure B.3 shows similar results for the rollout MSE. For most combinations of message-passing steps and number of solver iterations, C-GNS-GD (green) outperforms the Iterative GNN (yellow), C-GNS-FP (purple) as well as the Forward GNN (blue) with the same number of MPs (the Forward GNN is not iterative model, so we plot it as a single bar). We hypothesize that the solver iterations in the C-GNS and Iterative GNN may play a similar role to message passing with shared weights .

**Interpreting the learned constraints** To better understand the learned $f_C$ functions in the C-GNS-GD, Figure 5 visualizes the node-wise constraint values as a function of $Y$ (proposed average velocity) for different nodes in the ROPE dataset while holding the other nodes' proposed update $Y$ fixed.

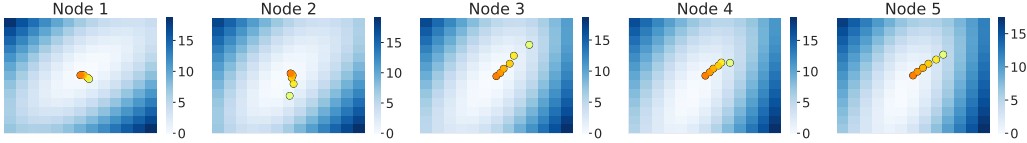

Figure 5: **Visualization of the constraint landscape for a trained C-GNS-GD.** Each subplot corresponds to a different ROPE node. The heatmap's color shows the constraint value evaluated at different values of $Y$ state. The plot is centered around the ground-truth point. The colored points show the five iterations of the constraint solver, from the initial $Y^{(0)}$ (yellow) to final $Y^{(5)}$ (orange).

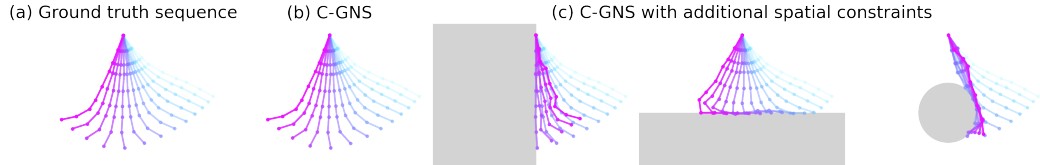

Figure 6: **Adding hand-designed constraints. (a)** The ground truth sequence of rope states, initialized at the cyan-colored state, and simulated over 14 time steps, to the final, purple-colored state. **(b)** The C-GNS-GD's rollout, without added constraints. **(c)** The C-GNS-GD's rollout, with wall, floor, and disk-shaped "forbidden" zones, imposed at test time via hand-designed constraint functions. A video of the trajectories is available at Videos.

We also overlay the sequence of five points that represent the proposed $Y^{(i)}$ steps from the solver where all nodes were jointly optimized. The figure shows the learned $f_C$ has a minimum near the ground truth Y, which the gradient descent steps are able to reach.

**Incorporating novel constraints at test time** We next explored a unique advantage of the constraint-based model: because the $f_C$ measures the degree the physical constraints are violated, we can incorporate additional, hand-designed constraints at test time, and use the model to potentially satisfy them. For the ROPE dataset, we designed three constraint functions that return positive values which increase quadratically as the rope enters different "forbidden" regions of the space: a vertical wall, a horizontal floor, and a disk-shaped region. We weighted these constraint terms by a coefficient hyperparameter and added each of the hand-designed constraints to the learned $f_C$ term of C-GNS-GD and ran the forward evaluation of the model.

As shown in Figure 6, the model was able to simulate the dynamics in a way that the corresponding forbidden region was avoided. In some cases, satisfying the joint constraint resulted in unintuitive behaviors, such as the rope links changing in length to adapt to the obstacle (Videos). However, this is to be expected, as the minimum of the joint constraint may not overlap with the minimum of the learned constraint, which is the one that would otherwise guarantee length preservation. For this example we added a further hand-designed constraint that incentivizes maintaining relative distances between nodes. In general this is a powerful example of how constraint-based models can generalize outside their training data, and solve both for the learned dynamics and arbitrary desired constraints.

**Generalizing to larger systems via increased solver iterations** In principle, iterative and constraint-based simulators should find more accurate solutions by increasing the number of solver iterations, $N$. We investigated whether the C-GNS-GD and Iterative GNN trained on ROPE could generalize from $N_{train} = 5$ on which they were trained, to $N_{test} \in [0, 15]$. We also analyzed whether increased solver iterations could improve generalize performance from training on ropes with $5 - 10$ nodes, to test ropes with 20 nodes.

Figure 7a (top row) shows that for test ropes that match the $5-10$ nodes experienced during training, the Iterative GNN (light blue) overfits very heavily to $N_{test} = N_{train} = 5$: error increases abruptly for $N \leq 4$ and $N \geq 6$. By contrast, the C-GNS-GD (light red) generalizes much better to different $N_{test}$. Figure 7a (bottom row) shows that for test ropes with 20 nodes, the Iterative GNN again overfits, while the C-GNS-GD can generalize well to longer ropes if $N_{test}$ is increased.

We also trained the Iterative GNN and C-GNS-GD with additional loss terms that were applied to the $Y^{(i)}$ on each solver iteration, not only the final one, $\hat{Y} = Y^{(N)}$. We used an exponential decay factor, $\alpha = 0.25$, which downweighted this additional loss term more heavily for earlier solver proposals. The dark blue and red curves in Figure 7a show how this additional loss further improves generalization to more solver iterations and larger systems as test time for the Iterative GNN, but especially the C-GNS-GD. Figure 7b visualizes how increasing the solver iterations systematically improves the quality of the long-term rollout accuracy in the ROPE dataset.

Together these results show the C-GNS-GD is effective in making use of additional resources at test time. This opens the exciting possibility of training on small, simple systems, and testing on large, complex systems. See Supplementary Figure B.2 for further details.

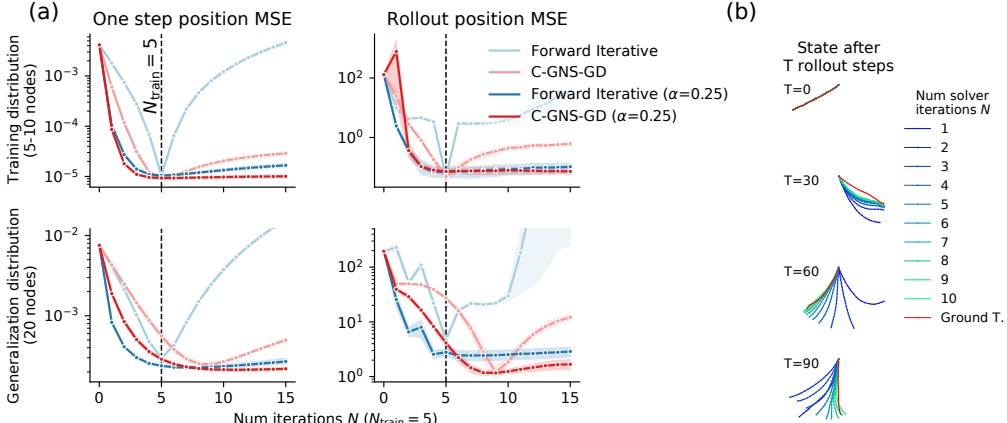

Figure 7: **Generalization to more solver iterations and larger ROPE systems at test time. (a)** Top row: test accuracy for ropes with the same lengths as those during training (5-10 nodes). Bottom row: test accuracy for larger ropes (20 nodes) than during training. Left column: 1-step MSE. Right column: full 160-step rollout MSE. The x-axes indicate the number of solver iterations at test time (training used 5 iterations). The y-axis represents MSE values. Different line colors represent different models, as indicated by the legend. **(b)** Example of the rollouts from C-GNS-GD with different number of solver iterations.

## 6 DISCUSSION

We presented a general-purpose framework for constraint-based learned simulation, where a learned constraint function implicitly represents the dynamics, and future predictions are generated via a constraint solver. We implemented our framework using GNNs as the constraint function and gradient descent as the constraint solver, and tested it in a variety of challenging physical simulation problems. Our results showed that our C-GNS has competitive or better performance compared to previous learned simulators. We demonstrated unique abilities to generalize to novel, hand-designed constraints, and use more solver iterations than experienced during training to improve the accuracy on larger systems.

We can hypothesize about the relationship between explicit, forward learned simulators and implicit, constraint-based ones in terms of the sharing schemes of these architectures. The C-GNS has a stronger inductive bias than the Forward GNN. The transformation of $f_C$ in C-GNS effectively ties the parameters in the resulting $\nabla_Y f_C$ function, and the solver iterations are analogous to how a recurrent neural network's parameters are shared over iterations. In contrast, the message-passing steps in the Forward GNN used in our work are unshared. In principle, the $f_D$ of the Forward GNN is more expressive because if given enough depth, after training it could learn to take parameter values that are equivalent to the shared parameters of C-GNS. Our results shown in Figure 4 supports this possibility: the Forward GNN with many more message-passing steps eventually approaches the C-GNS's performance. Moreover, we speculate the C-GNS's inductive biases contribute to its advantages in terms of incorporating novel hand-designed constraints and generalizing to more solver iterations and larger systems.

More broadly, the performance, generality and unique advantages of constraint-based learned simulation make it an important new direction in the advancement of machine learning methods for complex simulation problems in science and engineering.

## 7 REPRODUCIBILITY STATEMENT

We are committed to open-source the model code after the paper is accepted. Also, we are going to open-source the MuJoCo datasets that we generated for this paper. We provide more details on the model implementation as well as the hyperparameters used for each model in the Supplementary Material.

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

SUPPLEMENTARY MATERIAL

## A  IMPLEMENTATION

### A.1  THE DATASETS

We generate the ROPE, BOUNCING BALLS and BOUNCING RIGIDS datasets using MuJoCo physical simulator, using a MuJoCo timestep of 0.001 and taking every 30th time point. These datasets contain 8000/100/100 train/validation/test trajectories of 160 time points each.

**ROPE**  The rope is attached on one end and moves under the force of gravity in 2D space. We randomly sample the number of nodes in [5, 10] interval and the length of the rope links in [0.6, 1.1] interval. The length of the rope links remains constant during the simulation.

**BOUNCING BALLS**  The simulation of the balls bouncing inside the box in 2D. The number of bouncing balls is randomly sampled in [5, 10]. The radius of the ball is randomly sampled in [0.11, 0.3]. The size of the box is fixed to 5x5 in MuJoCo coordinates.

**BOUNCING RIGIDS**  A similar simulation to BOUNCING BALLS, where the nodes are connected into a 2D rigid structure that bounces inside the box. We randomly sample the number of nodes between 3 and 6.

**BOXBATH (Li et al., 2019)**  This is a 3D particle-based dynamics using the FleX engine of the fluid enclosed inside a box, with a rigid cube floating on the surface of the fluid. Each simulation contains 960 fluid particles and 64 particles representing the cube. The dataset contains 2700/10/100 training/validation/test trajectories with 150 time steps each.

We demonstrate the example of the rollouts for each environment in Supplementary Figure B.1 and Videos.

### A.2  CONSTRUCTING THE INPUT GRAPH

As mentioned, we append a history of three[6] recent velocities (see figure B.5(a-b) for ablations) for each node as node features. We also compute relative positions for each edge by subtracting the most recent positions in the history between the two particles adjacent to the edges, and use this as edge features.[7] Note we do not include the absolute positions as node features as the laws of physics are supposed to be invariant of the object's position in the space. We find that including the absolute positions into the nodes features harms the ability to generalize to a larger environment, such as a longer rope.

Note that we do not provide the 'rest shape' true pairwise distances between the nodes for the ROPE or rigid structures in BOUNCING RIGIDS and BOXBATH. This makes the task harder as the model has to infer the rest shape from the input history, and may gradually deviated from the true shape.

**Parameterizing the update $Y$**  For ROPE, BOUNCING BALLS, BOUNCING RIGIDS we use the average velocity as the update $Y$. For BOXBATH, the proposed update represents the normalized acceleration of the particle, initialized to zeros. We use normalized acceleration as the update $Y$, rather than raw velocity to better match the approach in Sanchez-Gonzalez et al. (2020); Pfaff et al. (2021). This proposed $Y$ is is also concatenated as an extra node feature at each optimization iteration, and optimized to match the target future normalized acceleration. This acceleration is then un-normalized, and then Euler-integrated twice to produce the next position. We chose to used normalized acceleration as the update $Y$, rather than raw velocity to better match the approach in Sanchez-Gonzalez et al. (2020); Pfaff et al. (2021).

---

[6]We use five for BOXBATH to match the model in Sanchez-Gonzalez et al. (2020)

[7]For BOXBATH, we also provide the norm of the relative distances as an additional vector to match features in Sanchez-Gonzalez et al. (2020)

**Handling walls**   Similar to Sanchez-Gonzalez et al. (2020), we include the euclidean distance between the center of the node to each of walls as additional node features, clipping it at a maximum value to avoid this to become a proxy for absolute position. For BOUNCING BALLS and BOUNCING RIGIDS we clip the distance at 2, and for BOXBATH at 0.08. For the iterative models, we update the distances to the walls after every step of constraint optimization.

**Node type**   We also provide an additional one-hot node feature indicating the node type (e.g. rigid, fluid, fixed). For BOUNCING BALLS, for which the object size varies between the objects, we provide the radius of the object as an additional node feature. Other features, like the relative positions are computed between centers of the nodes, and the network has to account for the object size to detect a collision.

**Graph edges**   In BOUNCING RIGIDS and BOUNCING BALLS we use a fully-connected graph. In ROPE we add edges between nodes that are adjacent within the rope. In BOXBATH we add edges between particles that are within a radius of 0.08 from within each other, and then recompute these edges at every step of a rollout according to the updated positions (as in Sanchez-Gonzalez et al. (2020)).

**Normalization**   For the large scale datasets BOXBATH we found it was important to normalize inputs and targets to zero-mean unit-variance (as in Sanchez-Gonzalez et al. (2020)). In the other datasets, BOUNCING BALLS, BOUNCING RIGIDS, and ROPE, the scale of the features was already close to zero-mean unit-variance, except for the input/target velocities in BOUNCING BALLS, BOUNCING RIGIDS to which we applied a scaling factor 100.

**Noise**   To stabilize rollouts in BOXBATH, we added noise to the input sequences in the same manner and with the same magnitude as in Sanchez-Gonzalez et al. (2020).

**Fixed particles**   Some of the datasets contain the fixed nodes that do change the their position in the simulation, like the first node in the ROPE. We prevent the update for those nodes by using *stop_gradient*.

## A.3   MODEL IMPLEMENTATION

**Computing the constraint gradients**   To compute the gradients of the constraint scalars for the batch of graphs, we use the vector-jacobian product (vjp) function. VJP does not explicitly construct a jacobian, and its asymptotic computational cost is the same as the forward evaluation of the constraint function.

**Constraint function**   We use the mean aggregation for the per-node outputs to obtain the scalar constaint value for the entire graph. For gradient descent, we additionally take a square of per-node outputs before aggregating them. We use a fixed learning rate of 0.001 for Gradient Descent constraint solver.

**ConstraintMLP**   For the ConstraintMLP we use a similar setup to as in the original paper by Yang et al. (2020). We concatenate the features for each node into a single vector and run an MLP to produce a scalar constraint output. The node features include the absolute positions, velocities and distances to the walls for each node for the optimized state and for the past three time points. Note that this model does not have access to the position differences between the nodes (this information is in the edge features for the graph network). Therefore we add the absolute positions to the node features, which simplifies the task for the model. Additionally, we adapt the ConstraintMLP to handle the scenes with the variable number of the nodes. To do so, we pad the missing nodes with zeros vectors up to the maximum number of nodes in the dataset.

## A.4   HYPERPARAMETERS

**Rope**   We used 2 message-passing step for the graph-network-based models. The MLPs for node and edge processing consist have 3 hidden layers with 256 hidden units, and hidden node and edge latent size of 64. We use 'softplus' activation and a LayerNorm Ba et al. (2016).

For the ConstraintMLP, we performed the hyperparameter search and chose the best performing configuration. We use the MLP with 10 hidden layers and 2048 hidden units with 'softplus' activation with no LayerNorm for the constraint function. We include the absolute positions into the node features, otherwise the ConstraintMLP model does not contain any information about the relative positions of the nodes (that we add into the edges for the graph model).

**Bouncing Balls**    We used 1 message-passing step for the graph-network-based models. The MLPs for node and edge processing consist have 3 hidden layers with 256 hidden units, and hidden node and edge latent size of 64. We use 'softplus' activation and LayerNorm after every MLP, except the final decoder.

For ConstraintMLP we used the best-performing configuration from our hyperparamter search. We use the MLP with 10 hidden layers and 2048 hidden units with 'softplus' activation with no Layer-Norm.

**Bouncing Rigids**    We used 1 message-passing step for the graph-network-based models. The MLPs for node and edge processing consist have 3 hidden layers with 256 hidden units, and hidden node and edge latent size of 64. We use 'tanh' activation and LayerNorm after every MLP, except the final decoder.

For ConstraintMLP, we use the MLP with 10 hidden layers and 2048 hidden units with 'softplus' activation. On this dataset, we found that adding a LayerNorm to ConstraintMLP-FP model helps the stability of the model.

**Box Bath**    We used 1 message-passing step for the graph-network-based models. All other hyperparameters are as in Sanchez-Gonzalez et al. (2020). The MLPs for node and edge processing consist have 2 hidden layers with 128 hidden units, and hidden node and edge latent size of 128. We use 'softplus' activation and LayerNorm after every MLP, except the final decoder.

**Training**    We train the models for 1 million steps on ROPE, BOUNCING BALLS and BOUNCING RIGIDS. We used the initial learning rate of 0.0001, with the decay factor of 0.7 and learning schedule of (1e5, 2e5, 4e5, 8e5). We use a batch size of 64. We trained for 2.5 million steps for the experiments studying the number of solver iterations. On BOX BATH we trained for 2.5M steps with a batch size of 2. With a learning rate starting at 0.001 and decaying continuously at a rate of 0.1 every 1M steps, as in Sanchez-Gonzalez et al. (2020).

**Loss on multiple iterations**    For the variable solver iterations results we experimented training models with a loss imposed not only at the last iteration output $\hat{Y}^{(N)}$ but at all intermediate outputs $\hat{Y}^{(i)}$ of the iterative models. We used exponentially decaying relative weights (from last to first), such that the relative weight $w_i$ for the loss term for the $\hat{Y}^{(i)}$ output was $w_i = \alpha^{N-i}$. The goal of $\alpha$ is to encourage the model to make progress towards the minimum at each iteration, without penalizing the model heavily for not reaching the solution in $i < N$ iterations. More details on the choice of $\alpha$ are provided on Supp. Fig. B.2.

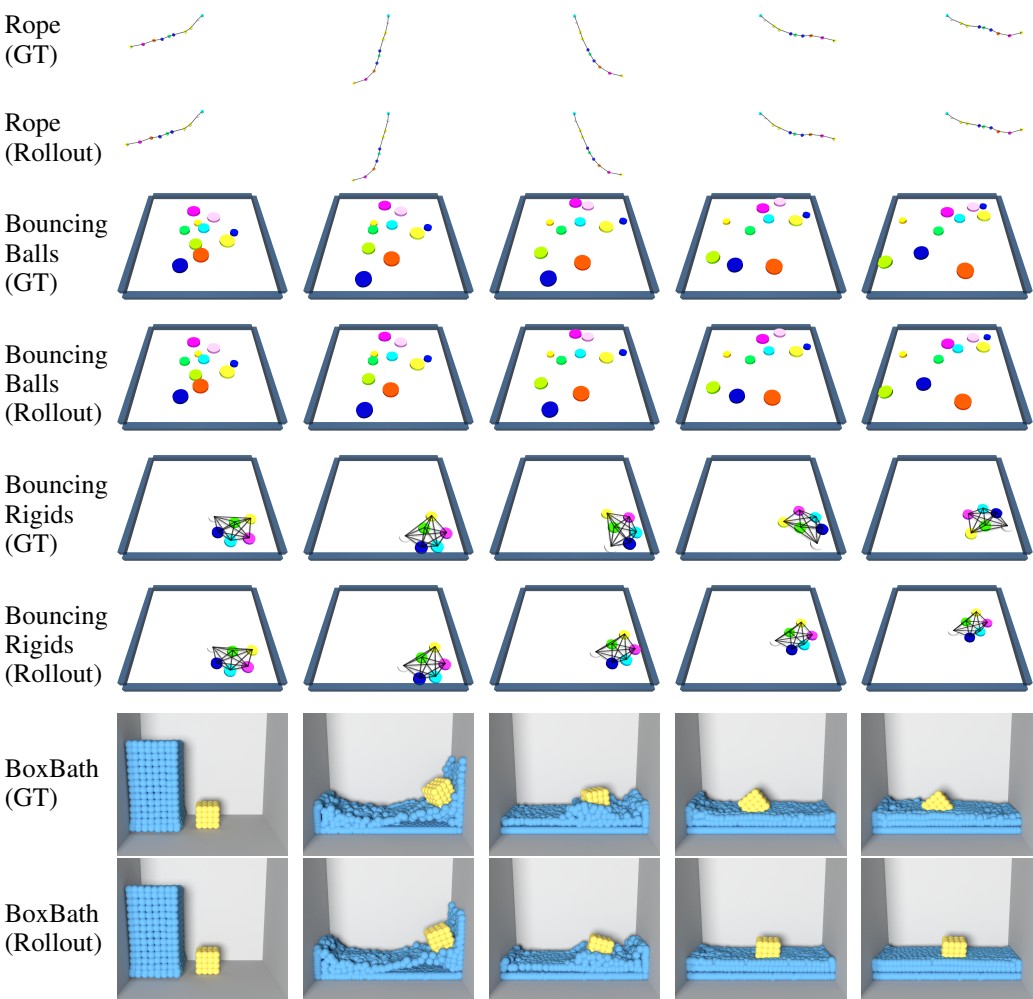

Figure B.1: Examples of the rollouts for our simulation environments.

## B SUPPLEMENTARY PLOTS AND TABLES

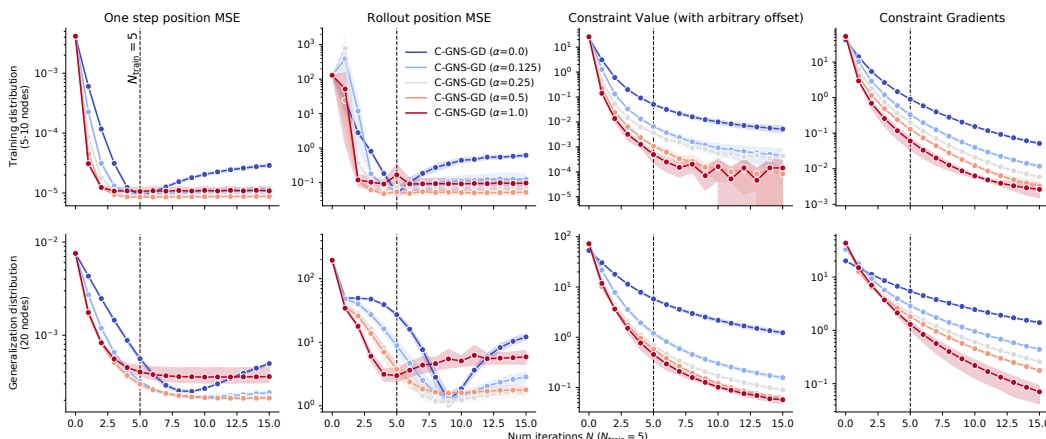

Figure B.2: Generalization to more solver iterations at test time $N$ as function of $\alpha$. Imposing loss only at the last iteration ($\alpha$=1), causes the model to

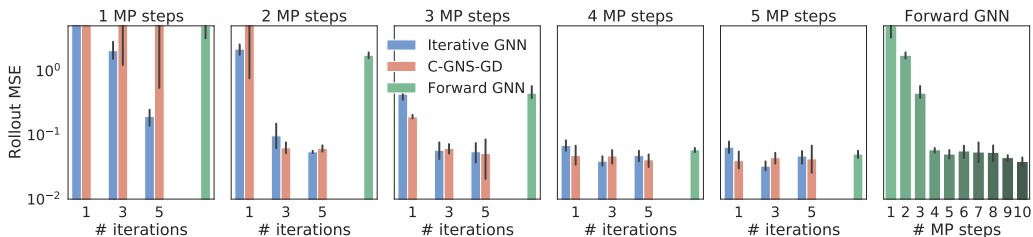

Figure B.3: Comparison of C-GNS to the baselines with different number of message-passing layers and number of constraint solver iterations (Test MSE on the full rollout). Last facet: 1-step rollout error of the state-of-the-art Forward GNN with 1 to 10 message-passing steps for reference.

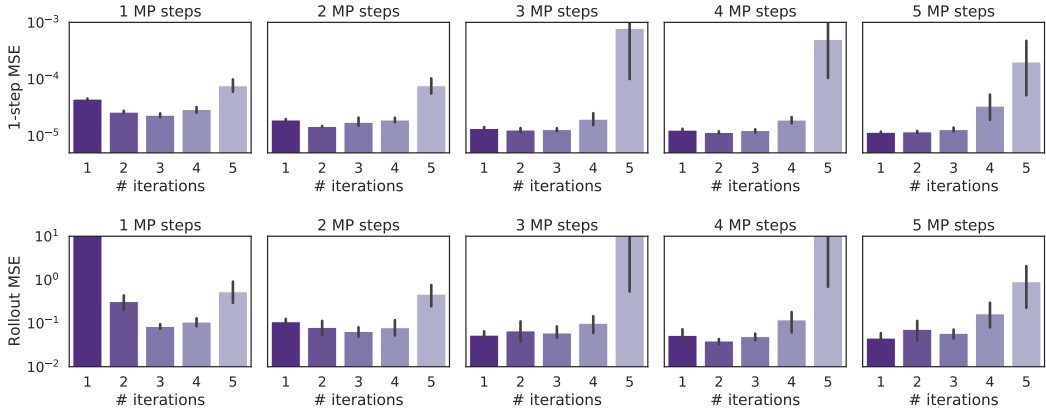

Figure B.4: Test 1-step MSE and Full Rollout MSE of the C-GNS with Neural Projection with different number of message-passing layers and number of constraint solver iterations on the Rope dataset. Top row: position MSE on the full rollout.

Table B.1: Median performance of the models on different datasets. ***ConstraintMLP**: results not shown for the 'BoxBath' dataset as it is infeasible to run MLP constraint on 1000s of nodes. The standard deviation from the median is shown over 5 random seeds. We do not show the results for the models where the median value, or standard deviation is more than 1000 times larger than the best model in each column. Results for all seeds are shown in figure B.6 Note that the tables use different scales to demonstrate the errors on 1-step error, 10-step rollouts and full rollouts.

| Model | One-step position MSE | | | |
|---|---|---|---|---|
| | Rope (1e-4) | Bouncing Balls (1e-6) | Bouncing Rigids (1e-7) | Box Bath (1e-7) |
| ConstraintMLP-GD* | $0.740 \pm 0.021$ | $7.716 \pm 1.610$ | $8.047 \pm 1.029$ | – |
| ConstraintMLP-FP* | $32.332 \pm 90.408$ | $31.336 \pm 0.000$ | $25.221 \pm 0.000$ | – |
| Forward GNN | $0.406 \pm 0.005$ | $0.126 \pm 0.009$ | $1.152 \pm 0.051$ | $1.440 \pm 0.038$ |
| Iterative GNN | $0.135 \pm 0.001$ | $0.168 \pm 0.019$ | $1.172 \pm 0.298$ | $1.042 \pm 0.103$ |
| **C-GNS-FP** | $0.698 \pm 0.297$ | $5.896 \pm 5.930$ | – | $0.954 \pm 1.885$ |
| **C-GNS-GD** | $\mathbf{0.110 \pm 0.005}$ | $\mathbf{0.103 \pm 0.032}$ | $\mathbf{0.884 \pm 0.163}$ | $\mathbf{0.998 \pm 0.038}$ |

| Model | Rollout position MSE (10 steps) | | | |
|---|---|---|---|---|
| | Rope (1e-3) | Bouncing Balls (1e-5) | Bouncing Rigids (1e-4) | Box Bath (1e-5) |
| ConstraintMLP-GD* | $1.463 \pm 0.231$ | – | $0.998 \pm 0.436$ | – |
| ConstraintMLP-FP* | – | $692.940 \pm 0.000$ | $4.460 \pm 0.000$ | – |
| Forward GNN | $27.585 \pm 7.335$ | $2.853 \pm 0.277$ | $1.217 \pm 0.346$ | $0.386 \pm 0.061$ |
| Iterative GNN | $\mathbf{0.251 \pm 0.145}$ | $2.260 \pm 0.429$ | $\mathbf{0.238 \pm 0.026}$ | $\mathbf{0.174 \pm 0.020}$ |
| **C-GNS-FP** | $4.458 \pm 2.323$ | $661.72 \pm 372.49$ | – | $0.200 \pm 0.321$ |
| **C-GNS-GD** | $\mathbf{0.226 \pm 0.058}$ | $\mathbf{0.613 \pm 0.333}$ | $\mathbf{0.248 \pm 0.038}$ | $0.288 \pm 0.055$ |

| Model | Rollout position MSE | | | |
|---|---|---|---|---|
| | Rope (1e-1) | Bouncing Balls | Bouncing Rigids (1e-1) | Box Bath (1e-2) |
| ConstraintMLP-GD* | $20.619 \pm 94.212$ | $2.161 \pm 0.217$ | $4.241 \pm 1.970$ | – |
| ConstraintMLP-FP* | – | $6.664 \pm 0.000$ | $13.128 \pm 0.000$ | – |
| Forward GNN | $18.305 \pm 2.340$ | $0.389 \pm 0.022$ | $1.174 \pm 0.486$ | $0.756 \pm 0.089$ |
| Iterative GNN | $\mathbf{0.546 \pm 0.026}$ | $0.445 \pm 0.044$ | $\mathbf{0.306 \pm 0.253}$ | $\mathbf{0.609 \pm 0.015}$ |
| **C-GNS-FP** | $2.804 \pm 3.432$ | – | – | $0.689 \pm 0.099$ |
| **C-GNS-GD** | $0.602 \pm 0.088$ | $\mathbf{0.308 \pm 0.142}$ | $0.374 \pm 0.171$ | $0.654 \pm 0.002$ |

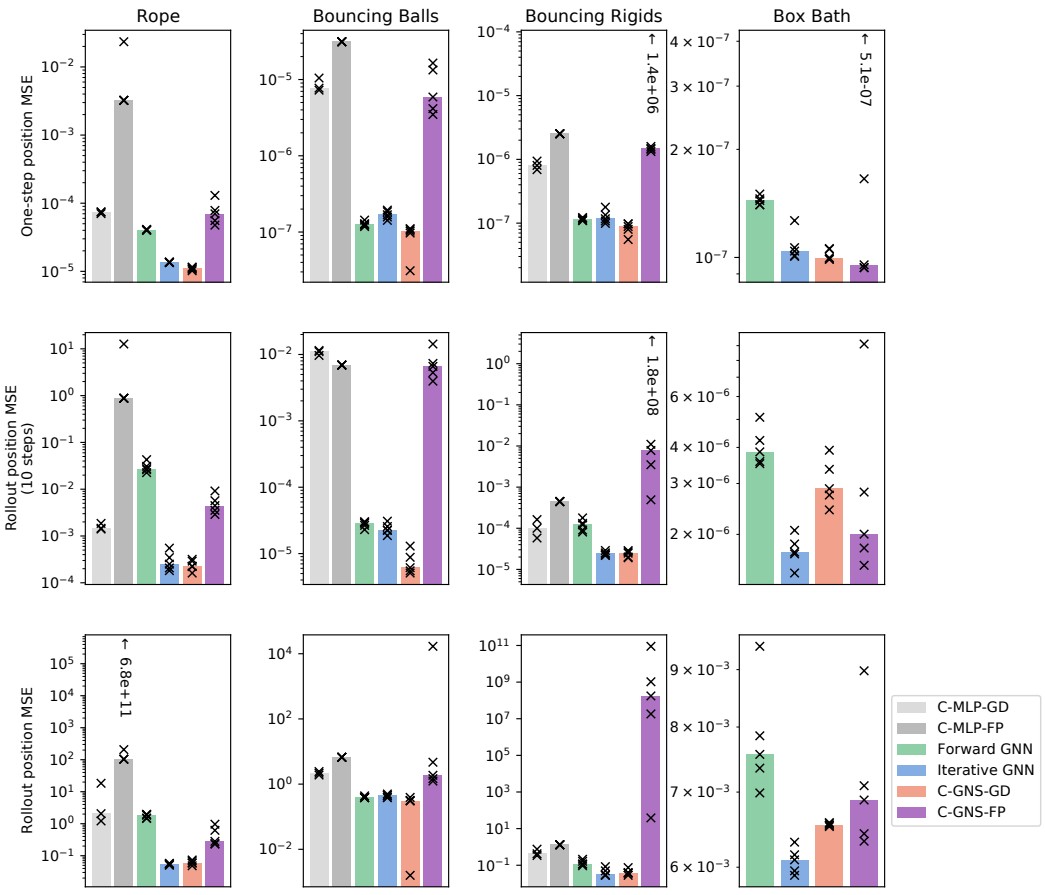

Figure B.6: **Test MSE on the node positions predictions across different models.** First row: 1-step position MSE. Second row: position MSE error for the first 10 steps of a rollout. Third row: position MSE error full 160-step rollout MSE. The bar height represents the median MSEs over random seeds. The × marks show the MSE metric for each random seed. The black arrows indicate the MSE metric for a random seed exceeding the upper y limit of the figure.

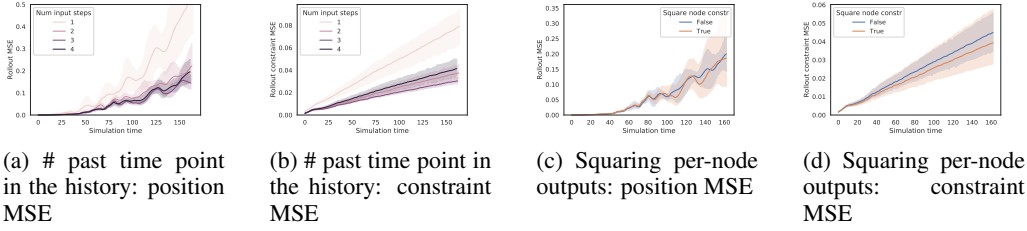

(a) # past time point in the history: position MSE

(b) # past time point in the history: constraint MSE

(c) Squaring per-node outputs: position MSE

(d) Squaring per-node outputs: constraint MSE

Figure B.5: Ablations of the modelling choices.

