# OpenReview forum: "Constraint-based graph network simulator"
_ICLR.cc/2022/Conference — ICLR 2022 Submitted_

### Official Review · Reviewer_TMn2 · 2021-10-28

**Correctness:** 3
**Technical Novelty And Significance:** 2
**Empirical Novelty And Significance:** 2
**Recommendation:** 5
**Confidence:** 3

**Details Of Ethics Concerns:**

None.

**Main Review:**

1. I find the notations X and Y with and without hats very confusing from the beginning. It would be great if the paper could clarify the semantic meaning of these notations.
* Fig. 1 (a): its caption says the UPDATER uses $\hat{Y}$ to update $X_t$, but the figure shows an UPDATER using $Y$ to update $\hat{X}\_t$. The notations in Fig. 1 (a) itself is extremely confusing: it starts with $X_1$ (without a hat), proceeds to $\hat{X}\_t$ (with a hat), but then uses $X\_{\leq t}$ (without a hat), and finally predicts $\hat{X}_{t+1}$ (with a hat). How did you determine when to use a hat and when not?
* When defining the rollout, $X$ with and without hats are mixed. From the rollout’s definition, it starts with $X_t$ but is followed by $\hat{X}\_{t+1}$, $\hat{X}\_{t+2}$, …. . Does this mean you manually choose t and separate the whole rollout into two parts: before $t$ every $X$ is without a hat and after $t$ every $X$ is with a hat (generated by the simulator)?

2. I do not see a strong reason for introducing the explicit iterative simulator for comparison. It looks like $f\_{DI}$ can be rewritten as $f_D$ plus a few linear operators: $f\_{DI}(X_{\leq t}) = (f_D(X\_{\leq t}) - X_t) / N$. Therefore, I would expect $f_D$ and $f\_{DI}$ to have very similar capabilities. What insight would we expect to get from comparing them?

3. It looks like the proposed constraint-based predictor runs a fixed number of gradient-based iterations and a fixed step size without guaranteeing the constraint is satisfied eventually. Therefore, I feel it is a bit too much to claim that the proposed approach is doing constraint-based simulation and to claim that the proposed network is a constraint solver. I think it would be necessary to either tone down this claim in the title, abstract, introduction, etc., or state explicitly at the beginning of the paper that the proposed approach does not guarantee the constraints are always satisfied.

4. The network design ensures translation-invariance by taking as inputs position offsets instead of absolute positions. Does the network also ensure rotation-invariance?

5. “These node-wise constraint values are averaged to compute a single scalar c constraint for the entire system”. This seems questionable to me if the goal is to solve constraints $f_C(X\_{\leq t},  \hat{Y}) = 0$, in which case I would expect that using an average of $|c|$ or $c^2$ makes more sense.

6. Looking at the experimental task domains, I feel there are two types of constraints involved in the simulator: equality constraints (e.g., end points of the consecutive rope segments must share the same location) and inequality constraints (e.g., bouncing balls must have nonnegative distances to the boundary). Does this paper deal with both equality and inequality constraints?

7. Both N and lambda seem to be crucial hyperparameters that need to be chosen for different environments individually. If this approach is applied to a new environment, how would you determine a proper N and lambda?

8. “In principle, iterative and constraint-based simulators should find more accurate solutions by increasing the number of solver iterations, N.” I am not sure I fully agree with this claim because the iterative solver is not equipped with a line search algorithm that adaptively changes the step size.

9. The large rollout MSE (orders of magnitude larger than #timesteps x One-step MSE) in Fig. 3 bottom seems to imply the learned simulator is not a good replacement of a numeric simulator because it accumulates errors from all time steps. I understand this may be a common issue that many other neural-network-based simulators also suffer from (all baselines in Fig. 3 have accumulated substantial errors and produced large rollout MSE), but I am still wondering whether the authors could give people a strong reason why it is useful to develop such a neural-network-based simulator if it is not accurate.

10. Similarly, I wonder if this paper could provide more discussions on the generalizability of the learned network on the static properties of the environment (the Z vector in the main paper), e.g., density, material types, time step size, etc. My understanding is that the network needs to be retrained if Z is updated, or the training set needs to be augmented to see various Z values. This does not seem to be very ideal for a simulator. Again, I understand this may be a common problem for many neural-network-based simulation papers, so I won’t hold it against this paper too much.


**Summary Of The Paper:**

This paper presents a neural-network simulator that learns to solve constraints inspired by classic physics-based simulation. The proposed simulator uses a graph neural network to encode a constraint solver and shows results in a number of simulation environments.

The main contribution of the paper is its idea of using graph networks to encode constraint-based simulation.


**Summary Of The Review:**

I recommend rejection based on the current status of the paper. My concerns are listed above.

---

> ### Author Response · Authors · 2021-11-17
>
>
> **Similarly, I wonder if this paper could provide more discussions on the generalizability of the learned network on the static properties of the environment (the Z vector in the main paper), e.g., density, material types, time step size, etc. … Again, I understand this may be a common problem for many neural-network-based simulation papers, so I won’t hold it against this paper too much.**
>
> This is an excellent point and a fundamental question about learning simulation. Solving the full problem of generalization is slightly out of scope for this paper, however, we believe this paper is actually advancing in that direction.
>
> In Figure 6, we show that we can add manual constraints that were not present in the training dataset, and show that can be combined with the learned model to produce trajectories without any fine-tuning. One could imagine a similar approach to enforce behaviors for new materials never seen at train time. More generally, our model can also be combined with other approaches to improve generalization by embedding additional domain-specific inductive biases in the model (e.g. use it within a NeuralODE (Chen et al., 2019) type of model to be able to generalize to different values of the time step size).
>
> Next, graph-network-based models are able to train on small systems and generalize to larger systems, which is key for the applicability to real world problems. In this work we show that we can leverage the unique structure of the constraint-based model to improve the generalization accuracy to a larger system at test time, better than any of the baselines.
>
> We will add a discussion on generalization and emphasize our model contributions (and limitations) in that direction more clearly.

---

> > ### Comment · Reviewer_TMn2 · 2021-11-28
> > **Thank you for the responses**
> >
> > Thank you for your clarification. I have read all your responses above, and I appreciate your time and effort in improving this work. While I am still leaning towards rejection, I will raise my score from 3 to 5.

---

> ### Author Response · Authors · 2021-11-17
>
>
> **The large rollout MSE in Fig. 3 bottom seems to imply the learned simulator is not a good replacement of a numeric simulator because it accumulates errors from all time steps. I understand this may be a common issue that many other neural-network-based simulators also suffer from ..., but I am still wondering whether the authors could give people a strong reason why it is useful to develop such a neural-network-based simulator if it is not accurate.**
>
> This is a great question. We agree we did not sufficiently discuss this in this paper (as many of the baselines already argue these reasons), but we will make sure to include some of this motivation.
>
> - The first reason is, the classic **non-learned simulators typically require a significant human effort to build**. This process involves writing down the partial differential equations (PDE) for a particular domain and building a solver that can produce a sufficiently accurate solution and avoid numerical instabilities. For the new domain, these simulators have to be built from scratch. In contrast, learned simulators learn directly from data and, with a good training set, can achieve comparable accuracy to the classic simulators without as much human involvement.
>
> - The second reason is **domain independence**: a single model architecture can be trained and reused across domains. This may be critical in some domains where there may not be exact PDE solvers available, and the approximate solvers may not be good enough. In this case learning from data may be critical, and may lead to models that are more accurate than the best available numerical solver.
>
> - The third reason is **efficiency**: several recent papers (e.g. Pfaff et. al., 2020, Kochkov et. al 2021) have shown that it is possible to leverage spatial and temporal coarsening to train learned models that are faster than their numerical counterparts. While it is true that the error compared to the ground truth solver (ran at very high resolution and with very small time steps) is not exactly zero, it has been shown that in some cases it is possible to achieve a better compromise between resolution and speed, compared to, for example, running the numerical solver at coarser resolutions, as learned models can learn to compensate for discretization errors. Also, even for a domain where a good numerical solver exists, a hardware (GPU, TPU) accelerated implementation may not be available. In contrast, neural networks are optimized for running on modern hardware.
>
> - Last but not least, learned models are also by default **differentiable**, which is not the case for many numerical solvers. Having access to gradients of the model has important implications for design optimization.

---

> ### Author Response · Authors · 2021-11-17
>
> **The network design ensures translation-invariance by taking as inputs position offsets instead of absolute positions. Does the network also ensure rotation-invariance?**
>
> As reviewers pointed out, we ensure translation equivariance by provisioning only positional differences between the nodes, but the model is not rotation equivariant. We opted for a  non-rotation-equivariant model as this simplified comparison with the most relevant state of the art baselines for this work, which are also not rotational equivariant, and because the dataset we are using for most of the comparisons (Rope) breaks the rotational symmetry due to gravity.
>
> In principle, we can use any GNN model to parameterize the constraint, including the rotation-equivariant models from the recent literature.  Note that enforcing rotation equivariance is a hard problem on its own (see, for example, SE(3) graph network [1], DimeNet [2])
>
> [1] SE(3)-Equivariant Graph Neural Networks for Data-Efficient and Accurate Interatomic Potentials. S. Batzner, A. Musaelian, L. Sun, M. Geiger, J.P. Mailoa, M. Kornbluth, N. Molinari, Tess E. Smidt, B. Kozinsky, 2021
>
> [2] Directional Message Passing for Molecular Graphs. J. Klicpera, J. Groß, S. Günnemann, 2020
>
> &nbsp;
>
> **“These node-wise constraint values are averaged to compute a single scalar c constraint for the entire system”. This seems questionable to me if the goal is to solve constraints , in which case I would expect that using an average of  |c| or c^2 makes more sense.**
>
> This is absolutely right. In the case of the Gradient-Descent model, we additionally square the constraint output (c^2) to give an inductive bias that the minimum exists (section 4.3 ‘C-GNS Gradient Descent’ model). We will emphasize it more in the text.
>
> &nbsp;
>
> **… There are two types of constraints involved in the simulator: equality constraints (e.g., end points of the consecutive rope segments must share the same location) and inequality constraints (e.g., bouncing balls must have nonnegative distances to the boundary). Does this paper deal with both equality and inequality constraints?**
>
> This is a great question. It is worth distinguishing between the real physical constraints and the learned constraints that we are optimizing. We train the learned constraints such that they can be optimized by a gradient descent and the minimum is the correct future state of the system. Thanks to this distinction, our model is agnostic to the form of the real physical constraints, and, as the reviewer pointed out, we are able to model a variety of real physical constraints (such as equality or inequality) using a single model architecture.
> We understand  that we did not explain the distinction between the learned versus real physical constraints. We will make sure to explain this more in the text.
>
> &nbsp;
>
> **Both N and lambda seem to be crucial hyperparameters that need to be chosen for different environments individually. If this approach is applied to a new environment, how would you determine a proper N and lambda?**
>
> Yes, these parameters can definitely be tuned, if needed, to push the performance further.
> We found that the model is relatively insensitive to the choice of N (with N>=3) and the choice of lambda (we tried lambdas between 1e-5 to 1), so we opted for a fixed values of N and lambda for all our environments for the sake of consistency.
> We think that the reason for this insensitivity is the following. The actual step that the gradient descent takes will depend on the norm of the constraint gradient. The model fully defines the constraint and therefore can adjust the scale of the gradient norm according to lambda and N.
>
> &nbsp;
>
> **“In principle, iterative and constraint-based simulators should find more accurate solutions by increasing the number of solver iterations, N.” I am not sure I fully agree with this claim because the iterative solver is not equipped with a line search algorithm that adaptively changes the step size.**
>
> Yes, using line search is reasonable to explore. However, as the constraint surface is learned, it will require a separate study to understand how the adaptive solvers affect the learning of the constraint. The constraint is learned such that it is possible to minimize it with the available solver (e.g. line search or fixed-step gradient descent). We suspect that with line search the constraint will have small gradient norms and will not represent a funnel as strongly as in figure 5.
> For gradient descent optimizer, here we referred to the fact that, under certain assumptions, it is guaranteed to converge with the fixed step size (https://www.stat.cmu.edu/~ryantibs/convexopt-F13/scribes/lec6.pdf). However, we agree with the reviewer that our learned constraint surface does not necessarily satisfy these assumptions. We will soften this claim in the paper.

---

> ### Author Response · Authors · 2021-11-17
>
>
> **I find the notations X and Y with and without hats very confusing from the beginning in Fig. 1 (a). … How did you determine when to use a hat and when not?**
>
> The hat notation is standard for the papers that use model rollouts, including Sanchez-Gonzalez et al and Yang et al.
>
> The X without a hat is considered known and fixed at the current time point. $\hat{X}$ is used for the prediction of the model that we are optimizing . As such, when we do the prediction for the time point t+1, the states $X_{\leq t}$ are fixed, and we optimize the prediction $\hat{X}\_{t+1}$
>
> Then, for the time point $t+2$, we optimize the  $\hat{X}\_{t+2}$. Now we can consider the previous prediction $\hat{X}\_{t+1}$ to be fixed and assign $X_{t+1} = \hat{X}\_{t+1}$. We agree that this is a notation overload.
>
> We agree that figure 1(a) should have $\hat{Y}$ instead of $Y$, for consistency with figures 1 (b-d)
>
> We are going to revise some of the notation and simplify it for the final manuscript.
>
> &nbsp;
>
> **When defining the rollout,  with and without hats are mixed. From the rollout’s definition, it starts with $X_t$ but is followed by $\hat{X}\_{t+1}$ ... . Does this mean you manually choose t and separate the whole rollout into two parts: before  $t$ every $X$ is without a hat and after  $t$ every $\hat{X}\_{t+1}$ is with a hat (generated by the simulator)?**
>
> We follow the notation from the previous works, and we agree that there is a notation overload.
>
> To generate the rollout, we need to start with a sequence of ground-truth state(s) $X_{\leq t}$. In our case, $X_{\leq t}$ contains three past time points, either from ground-truth or past predictions. Then we use the model to generate the subsequent predictions $\hat{X}\_{t+1}$, $\hat{X}\_{t+2}$ for the time points $t+1$, $t+2$, etc.
>
> We chose to provide three past time points as $X_{\leq t}$ by performing the ablation studies. Please see figure B.5 (a-b) for the ablation on the rope dataset .
>
> &nbsp;
>
> **I do not see a strong reason for introducing the explicit iterative simulator for comparison. It looks like fDI can be rewritten as fD  plus a few linear operators: fDI(X≤t)=(fD(X≤t)−Xt)/N. Therefore, I would expect fD  and fDI  to have very similar capabilities. What insight would we expect to get from comparing them?**
>
> Yes, we also found the results on the Direct Iteration model surprising. With fDI model we wanted to test the model that performs iterative refinement of the future state, but predicts the delta directly rather than through optimizing the constraint. It is a model in-between the forward GNN and the constraint-based model. We felt that it was the most transparent and comprehensive to include this baseline.
> Figure 4 shows that the performance of fDI (Iterative GNN) model improves if trained with more iterations, demonstrating that it is not equivalent to fD. We hypothesize that it is due to iterating on the same state $Y^{(i)}$ , as we are applying the same set of weights N times (N is the number of iterations). The forward fD model is equivalent to fDI only if N=1.
>
> &nbsp;
>
> **It looks like the proposed constraint-based predictor runs a fixed number of gradient-based iterations and a fixed step size without guaranteeing the constraint is satisfied eventually. Therefore, I feel it is a bit too much to claim that the proposed approach is doing constraint-based simulation.**
>
> We refer to our approach as constraint-based because we model the physical simulation by predicting the constraint value instead of the future state directly.
>
> This is a great observation that we, in fact, do not guarantee that we find the exact minimum of the constraint with the fixed step size and number of iterations. Figure 5 demonstrates that our learned constraint function takes a funnel-like shape, and the gradient descent finds a solution that is sufficiently close to the minimum. In fact, our experiments with more solver iterations at test time (figure 7) show that further approaching the minimum translates into more accurate predictions. Therefore, In the future research, it is definitely worth trying adaptive methods for finding the minimum and exploring how they will affect the learning of the constraint. Note that not all classical solvers solve constraints to convergence either; it's e.g. quite common for PBD or rigid game engine solvers to use a fixed (or maximum) number of solver iterations.
>
> Importantly, we also do not claim that we guarantee the satisfaction of real physical constraints, as most neural-network-based approaches (including ours) provide merely an approximation for real physics. We will clarify this point in the manuscript as well.

---

### Official Review · Reviewer_WhyX · 2021-11-02

**Correctness:** 4
**Technical Novelty And Significance:** 2
**Empirical Novelty And Significance:** 3
**Recommendation:** 5
**Confidence:** 4

**Main Review:**

This manuscript extends an idea from [1] that the learning of physical simulation can be viewed as a constrained optimization problem. There are a few interesting points:

1. The authors extend it using graph neural networks. This modification generalizes this method to the indefinite number of states in the physical system.
2. The authors claim that this method trade-off between inference time and accuracy dynamically so that in the test time they are possible to increase the number of steps for better convergence.

Despite these respectful contributions, I still have a few questions and reservations:

1. I notice different activation functions are applied to different environments. Are they intentional? Is there a guideline for choosing?
2. LayerNorm will be affected by the magnitude of the static information. For example, if you simulate under the continuous mechanism, Young's Modulus can be so huge that kills other entries in the vector. I wonder how should the user deal with it.
3. How does this method generalize to the static information and time step?
4. Second-order optimization usually provides super-linear convergence when it is close to the optimum, which is a similar case referring to figure 5. I will find a comparison between the proposed first-order method and Newton's or quasi-Newton's method helpful.
5. The authors claim that adding an $\alpha$ improves the generalizability to unseen parameters including articulation length and optimization steps. Is there a reason why it is not a part of the standard model given it has an advantage?
6. 3 and 5 are separately used in different environments. Some discussion on the choice will be helpful (I understand 5 is from previous work). Is it because of the time integration method used in Mujoco?
7. My major reservation is the technical novelty compared to its preceding work [1]. I find the extend to graph neural networks exciting, but in the meantime, incremental. I wonder if the authors can give me more explanation on the delta between them, maybe supported with a more impressive application.

[1] Yang, Shuqi, Xingzhe He, and Bo Zhu. "Learning Physical Constraints with Neural Projections." Advances in Neural Information Processing Systems 33 (2020): 5178-5189.

**Summary Of The Paper:**

This manuscript proposes to learn the numerical solutions to Lagrangian physical simulation with a constraint-based inference method on graph neural networks. This method involves an iterative update during inference and thus enables test-time dynamical correction. The contributions are: (1) this manuscript builds a scalar predictor to indicate how well the constraint is agreed. (2) this manuscript proposes using the graph neural networks as the backbone to deal with a variable length of the physical domain. They also examine the effectiveness with a bunch of experiments including the state prediction experiment on four different environments and multiple ablation studies towards different hyper-parameters.

**Summary Of The Review:**

I appreciate the idea of using graph neural networks with a constraint-based learning paradigm. However, there are also a few technical questions remaining unsolved. I personally find this work is not satisfying me with solid novelty, but I am open to change if the authors address my concerns.

---

> ### Author Response · Authors · 2021-11-17
>
> **I notice different activation functions are applied to different environments. Are they intentional? Is there a guideline for choosing?**
>
> We noticed that the choice of the activation function affected the Fast Projection method more than Gradient Descent. For each dataset, we chose the activation function for which Fast Projection algorithm (C-GNS-FP) was more stable (more random seeds converged). Then we used the same activation function for other models, including Gradient Descent (C-CNS-GD), as the choice of the activation function had little impact on the performance for these models.
>
> &nbsp;
>
> **LayerNorm will be affected by the magnitude of the static information. I wonder how should the user deal with it.**
>
> Yes, this can indeed be an issue if the input features have drastically different scales. A common trick to improve training (e.g. Sanchez-Gonzalez, et al. 2020) is to use dataset normalization by normalizing each of the input features and output targets of the neural networks to have zero mean and unit variance across the full dataset. In our case this was not necessary for our datasets (Rope, Bouncing Balls and Bouncing Rigids), but we found it improved performance on the BoxBath dataset. In some cases where the dynamic range of the features may be large (e.g. Young’s modulus example), it may be beneficial to apply the normalization to the log of the variable.
>
> &nbsp;
>
> **How does this method generalize to the static information and time step?**
>
> We trained all models with the same time step, therefore we do not expect this generalization. Note that Yang et al., Pfaff et al. and Sanchez-Gonzalez et al. also used the fixed time step and did not show this type of generalization.
> We expect the models to generalize to a different size of the box, as the walls are provided as a clipped distance from the ball center to the wall. We also expect it to generalize to a different ball sizes and different length of rope links as those parameters vary across simulations in the training set.
>
> &nbsp;
>
> **Second-order optimization usually provides super-linear convergence when it is close to the optimum, which is a similar case referring to figure 5. I will find a comparison between the proposed first-order method and Newton's or quasi-Newton's method helpful.**
>
> In principle, we can definitely use other methods for solving the learned constraint other than Gradient Descent, provided that they are differentiable. Here we have already shown the proof-of-concept that the solution can be defined as a minimum of a learned constraint. Investigating the effect of different forward solvers to find the minimum goes beyond the scope of our paper.
>
>
> &nbsp;
>
> **The authors claim that adding an α improves the generalizability to unseen parameters including articulation length and optimization steps. Is there a reason why it is not a part of the standard model given it has an advantage?**
>
> Yes, it is a good point. We could potentially add the loss with weights  α, similarly to the generalization experiments. We observed that the additional loss was not necessary in other experiments that did not involve the generalization to more iterations or a longer rope (see top row of Figure 7 – the performance with 5 iterations is the same with and without the extra loss). Also for the sake of the paper, we wanted to showcase the distinct advantages of the constraint-based approach alone. However, in practice, it would make sense to use the additional loss on every iteration with weights α to enjoy better generalization properties, as we have demonstrated in figure 7.
>
> &nbsp;
>
> **3 and 5 are separately used in different environments. Some discussion on the choice will be helpful (I understand 5 is from previous work). Is it because of the time integration method used in Mujoco?**
>
> Can the reviewer clarify which parameter they are referring to?

---

> ### Author Response · Authors · 2021-11-17
>
>
> **Technical novelty compared to its preceding work [1]. … I wonder if the authors can give me more explanation on the delta between them, maybe supported with a more impressive application.**
>
> We recognize that it’s crucial to make the delta explicit, especially wrt Yang et al. ([1]). We addressed some of this in our response to reviewer #2 (see above), and we’ll add further text as follows.
>
> Our most significant contributions compared to Yang et al. are the following.
> - We add the conditioning of the constraint function on the past states X<=t. The previous approach by Yang et al. conditioned the constraints only on the current point, making it impossible to correctly model elastic or partially elastic collisions, as shown in this schematic [png file](https://drive.google.com/file/d/1JwU9cH0ap2hwnjfFzz6QzWTo8zV_Yt5R/view?usp=sharing)
> - We replaced the constraint network with the graph network that made it possible to run the constraint models on domains with >1000 nodes (BoxBath) with a variable number of nodes. Yang et al.’s MLP model is not feasible to run on the large domains like BoxBath and has 1-2 orders of magnitude worse performance on smaller domains (see C-MLP-GD and C-MLP-FP in figure 3).
> - We replaced the zero-finding algorithm (Fast Projections from Yang et al.) by Gradient Descent (see the high variance across seeds and poor performance of C-GNS-FP and C-MLP-FP in figure 3).
> - We demonstrated the unique properties of the model such as using hand-designed constraints and ability to tune the accuracy of the solver at test time, as well as interpretation of what the constraint function is learning. None of these types of experiments were performed in Yang et al.
> - We introduce the Iterative GNN model that gives an insight into whether the performance increase is due to the repeated iterations versus using the constraints.

---

### Official Review · Reviewer_AXAA · 2021-11-03

**Correctness:** 3
**Technical Novelty And Significance:** 2
**Empirical Novelty And Significance:** 2
**Recommendation:** 3
**Confidence:** 4

**Main Review:**

Strengths:

1.	This paper is overall well written. The authors have clearly demonstrated the pipeline of the proposed strategy.

2.	Although the novelty of combining GNN with constraint projection is weak (see the weaknesses below), it is valuable to check if this method can outperform those typical forward approaches (such as the work by Sanchez-Gonzalez et al., 2020). The experimental evaluations generally serve this purpose.


Weaknesses:

1.	The biggest concern is that the novelty is weak. At its core, this paper applies the pipeline in (Sanchez-Gonzalez et al., 2020), including the strategy by first computing the predictor and then updater, and the usage of GNN for interaction modeling. The main difference is that for the predictor, it replaces the traditional forward prediction with the iterative gradient-based solver of the constraint approximator, which is interesting. Yet, this idea has been proposed by Yang et al. (2020) in terms of the iterative projections along the gradient direction of the constraint network. Although the authors have further augmented the input of the constraint function with Y to take the dynamics into account, this modification seems minor and a straightforward enhancement.

2.	Regarding the constraint satisfaction. The authors first predict the changes in position (Y) and then update the next state X_{t+1} via an Euler integrator. Even the prediction of Y is derived via the constraint solver, the constraint will be broken after the following updater from X_{t} to X_{t+1}, which is problematic to maintain the hard constraints (such as the case in Figure 6 (c) where the movement within walls and floors is forbidden). How does the proposed method tackle this issue? In the work by Yang et al. (2020), the authors use an opposite order by first updating X_{t} and then projecting the positions onto the constraint manifold, which is able to meet any kind of constraints.

3.	For the comments above, there are several important baselines that are not tested in the experiments: 1) using the method by Yang et al. (2020) but with the GNN projector; 2) first updating X_{t} and then predicting Y with other setting unchanged in the current framework; 3) augmenting the explicit simulators (both the iterative and non-iterative versions) with a regulation loss to enforce certain hard-crafted constraints such as the cases in Figure 6 (c).

Other comments:

1.	This paper is almost well organized, but there are still some confusions in the current version.
1.1 In introduction, the authors mention that both families of simulators (explicit forward vs implicit constraint-based). Is this statement discussed in previous papers? Is there any citation?
1.2 In related work, the authors state that the neural projection by Yang et al. (2020) is the first work that uses learned constraints. However, they also introduce that “Recent methods have been proposed for learning constraint functions and solving them in a model’s forward pass” such as “Deep Implicit Layers” and “Deep Equilibrium Models”. These two statements seem self-contradictory.
1.3 In section 4.3, the authors claim that in Figure 3 the C-GNS-GD’s performance was generally better than the other model variants, which is not true. In terms of the rollout MSE, Iterative GNN outperforms C-GNS-GD in three out of four cases. The authors are suggested to provide more explanations here.

2.	It is good that the proposed method is translation-invariance. Yet, besides this symmetry, there are other cases, such as rotation invariance/equivariance. This is important for improving the generalization ability of the simulator, given the fact that if we rotate the input states under a certain angle, the output changes in the same way. Have the authors taken this symmetry into account?


**Summary Of The Paper:**

This paper presents to simulate physics via a constraint-based approach instead of direct prediction. In particular, the authors first employ GNN taking as input the history positions and dynamics, to capture the interaction between different particles within the system, whose output is considered as the constraint satisfaction scalar. Then, the gradient with respect to the constraint function is applied as the update of the dynamics over a certain number of solver iterations. The experiments are conducted on a variety of challenging physical domains, including simulated ropes, bouncing balls, colliding irregular shapes and splashing fluids.

**Summary Of The Review:**

Overall, given the limited novelty and insufficient experimental evaluations, I initially suggest weak rejection.

---

> ### Author Response · Authors · 2021-11-17
>
> As a final note, the reviewer has made a number of errors in their assumptions and characterization of our method and paper, especially wrt to Yang et al. 2020.. We respectfully request the reviewer raise their rating in light of these facts.

---

> ### Author Response · Authors · 2021-11-17
>
>
> **In section 4.3, the authors claim that in Figure 3 the C-GNS-GD’s performance was generally better than the other model variants, which is not true. In terms of the rollout MSE, Iterative GNN outperforms C-GNS-GD in three out of four cases. The authors are suggested to provide more explanations here.**
>
> We respectfully disagree that our claim is “not true”. We recognize the relative performances do not portray a clear winner between the Iterative GNN and C-GNS-GD (both our contributions). We state “We generally found that the Iterative GNN was fairly competitive with the C-GNS-GD in overall performance and better than the Forward GNN“
>
> To clarify, a robust pattern was evident in the 1-step MSE, where the C-GNS-GD more clearly outperformed the Iterative GNN in all 4 tasks. In the rollout MSE, the Iterative GNN had slightly lower error over the C-GNS-GD in two tasks, and slightly higher error in a third, but we felt all three were quite comparable because those differences were within seed variance (the fourth task had a clear advantage for the Iterative GNN). This was why we expressed our results as we did.
>
> We’re happy to add text discussing the rollout performance of the Iterative GNN and clarify the phrasing around “generally better” in the sentence pointed by the reviewer. We also addressed a related comment from the first reviewer, so see that as well for details.
>
> &nbsp;
>
> **It is good that the proposed method is translation-invariance. Yet, besides this symmetry, there are other cases, such as rotation invariance/equivariance. … Have the authors taken this symmetry into account?**
>
> As reviewers pointed out, we ensure translation equivariance by provisioning only positional differences between the nodes, but the model is not rotation equivariant. We opted for a  non-rotation-equivariant model as this simplified comparison with the most relevant state of the art baselines for this work, which are also not rotational equivariant, and because the dataset we are using for most of the comparisons (Rope) breaks the rotational symmetry due to gravity.
>
> Nevertheless, we believe rotational equivariance is an important direction for future work in simulation, and in principle the C-GNS model can be easily extended to be rotational equivariant by replacing the internal GNNs by any of the rotational equivariant GNNs available in the recent literature. Note that enforcing rotation equivariance is a hard problem on its own (see, for example, SE(3) graph network [1], DimeNet [2]).
>
> [1] SE(3)-Equivariant Graph Neural Networks for Data-Efficient and Accurate Interatomic Potentials. S. Batzner, A. Musaelian, L. Sun, M. Geiger, J.P. Mailoa, M. Kornbluth, N. Molinari, Tess E. Smidt, B. Kozinsky, 2021
>
> [2] Directional Message Passing for Molecular Graphs. J. Klicpera, J. Groß, S. Günnemann, 2020

---

> ### Author Response · Authors · 2021-11-17
>
>
> **Other comments:**
>
> **In introduction, the authors mention that both families of simulators (explicit forward vs implicit constraint-based). Is this statement discussed in previous papers? Is there any citation?**
>
> Here we refer to the space of classic physical simulators that resolve a set of hard-coded physical constraints. Some simulators directly compute forces or impulses (explicit), while others find the next state by optimizing the constraints (implicit). The latter is typically applied to the systems with many hard constraints. See, e.g. (Macklin et al 2014) for example shows pure-constraint based implementations of a range of dynamical systems which are classically implemented using explicit force-based methods.
>
> In the space of deep learning, the distinction between explicit versus implicit models goes  beyond the physical simulation domain in the recent works on implicit neural network layers. In our Related Work section we discuss this (see paragraph beginning “Recent methods have been proposed for learning constraint functions and solving them in a model’s forward pass”). However we will strengthen the connection to the explicit vs implicit simulation theme in the main text to emphasize this better.
>
> &nbsp;
>
> **In related work, the authors state that the neural projection by Yang et al. (2020) is the first work that uses learned constraints. However, they also introduce that “Recent methods have been proposed for learning constraint functions and solving them in a model’s forward pass” such as “Deep Implicit Layers” and “Deep Equilibrium Models”. These two statements seem self-contradictory.**
>
>
> There’s no contradiction here:
> In the Related Work section we wrote “Recent methods have been proposed for learning constraint functions and solving them in a model’s forward pass (Duvenaud et al. (2020)’s “Deep Implicit Layers” tutorial is an excellent hands-on survey)”. The Deep Implicit Layers / DEQ work are examples of learning constraint functions within deep learning architectures, but not for learned simulation.
> In the immediately following paragraph we wrote: “Despite the popularity of constraint-based traditional simulators, only a single simulator which uses learned constraints has been reported (Yang et al., 2020).” We were pointing to Yang et al is a constraint-based model for physical simulations. We did not say that Yang et al “is the first work that uses learned constraints”.
> Note, the Yang et al. paper did not draw connections to the implicit layers work, so perhaps this was the source of confusion?

---

> ### Author Response · Authors · 2021-11-17
>
>
> **There are several important baselines that are not tested in the experiments: 1) using the method by Yang et al. (2020) but with the GNN projector; 2) first updating X_{t} and then predicting Y with other setting unchanged in the current framework; 3) augmenting the explicit simulators (both the iterative and non-iterative versions) with a regulation loss to enforce certain hard-crafted constraints such as the cases in Figure 6 (c).**
>
> 1) We did actually test this baseline, as well as improved variants. See page 4 where we wrote: “We also explore a second constraint-solving procedure, inspired by Yang et al. (2020)’s Neural Projections’ use of “fast projection” Goldenthal et al. (2007).“ As detailed in our above response, our C-GNS-FP variant is Yang et al’s model with a GNN project, and with the addition of using the input state to also condition the constraint function. We found that without this additional input state modification the Yang et al + GNN model did not perform as well as any methods we reported, and so we chose to not report this, though we’re happy to add them back in if requested.
>
>
> 2) We actually tested this in preliminary experiments and found that updating the $X_t$ before passing it to the solver did not affect the performance, so we did not bother reporting this ablation. As an aside, we also experimented with using a separate, dedicated GNN to first predict $X_t$ before passing it to the solver, but again did not find much different in performance. It’s important to keep in mind that in our framework, updating $X_t$ first only affects the initialization of the solver’s proposed state, so if the constraint function captures the dynamics well, we shouldn’t expect how the solver is initialized to impact the solution it finds. By contrast, Yang et al’s model would be more sensitive to how the solver is initialized, because the proposed state is the only information the solver has regarding the input state. The “Projection” in “Neural Projections” refers to the fact that the solution is projected back to the constraint manifold at a point that’s nearby the initial $X_t$ proposal. We view this as strictly a limitation/weakness of Yang et al compared to our approach: if our approach could benefit from only being aware of the $X_t$ proposal, as in Yang et al., the training process could simply tune the constraint function to ignore the additional current state that’s provided as input to the solver. Moreover, because Yang et al’s specific projection solver procedure carries the inductive bias that the solution be near the initial proposal, this makes the model sensitive to the accuracy of the proposals, which isn’t desirable.
>
>
> 3) We don’t regard that as an “important baseline” and we don’t think it is comparable to ours --it’s a different training regime, where the network is *trained* with access to the true constraint functions. In our paper, the constraints in Figure 6(c) were introduced at test time only. This suggestion assumes the training process has access to the underlying physical constraints that *generated the training data*.  While this suggestion makes sense in certain regimes, we wanted to demonstrate that, when the model doesn’t have knowledge of the constraint at training time, it can still be trained in such a way that can respect new constraints at test time. What if you want to train a model on a dataset of swinging ropes someone gives you, then at test time simulate what would happen if a rigid object were added to the system? Because you cannot control the data-generating process, you could not incorporate that regularization loss. Just to note, the regularization loss idea is closely related to how “physics-informed neural networks” (e.g. Raissi et al, 2019) work, and therefore, we’d expect it to work in our setting as well. However, we’re not sure how it would change our conclusions whatever the results show, so we don’t feel it’s necessary to include it.
>
> To summarize, we’ve already tested and reported on (1), we tested (2) but found little difference so didn’t feel it was important to report, and we believe (3) isn’t an “important baseline”, and wouldn’t change our conclusions regardless of the outcome.

---

> ### Author Response · Authors · 2021-11-17
>
>
> **The authors first predict the changes in position (Y) and then update the next state X_{t+1} via an Euler integrator. Even the prediction of Y is derived via the constraint solver, the constraint will be broken after the following updater from X_{t} to X_{t+1}, which is problematic to maintain the hard constraints (such as the case in Figure 6 (c) where the movement within walls and floors is forbidden). How does the proposed method tackle this issue? In the work by Yang et al. (2020), the authors use an opposite order by first updating X_{t} and then projecting the positions onto the constraint manifold, which is able to meet any kind of constraints.**
>
> This is incorrect. Our constraint was conditioned on both the current state that’s input to the learned simulator, as well as the proposed state. This means the solver is always informed about the past state across all solver steps. In principle, the constraint function could actually apply its proposed update to the input state, evaluate whether the proposal would break the constraint, and update the proposal so that doesn’t happen. Had we used an approach like in Yang et al, where the input state is not made available to the constraint solver, the reviewer’s concerns would have merit, but our approach is not subject to those ordering concerns.
>
> The reviewer’s comment that Yang et al. “is able to meet any kind of constraints” is incorrect for similar reasons as just outlined. Specifically, Yang et al.’s model cannot learn to meet any constraints that require evaluating two states across time. For example, it cannot model the time dynamics themselves as a constraint. Consider the cases shown in this diagram: [png file](https://drive.google.com/file/d/1JwU9cH0ap2hwnjfFzz6QzWTo8zV_Yt5R/view?usp=sharing). In the top scenario, the Euler step proposes the ball moves past the thin wall. Since the Yang et al’s constraint function doesn’t treat this as a constraint violation, the ball will continue along happily as if the wall doesn’t exist! In the bottom example, the Euler step places the ball within the wall. Because the nearest constraint-satisfying location for the ball is on the edge of the wall, their approach will choose that location as the solution, regardless of the ball’s initial position and velocity before the wall collision. Hence it cannot even model an elastic collision properly: multiple input states could be mapped to the same output state, which is incorrect. This is also part of the reason why we believe Yang et. al.’s model cannot learn external forces from data, and instead requires these to be hard-coded as part of the Euler update.
>
> Generally Yang et al.’s model cannot learn a constraint defined over time such as energy preservation: once the Euler step breaks energy preservation, the proposed future state does not contain enough information about the energy of the previous state to be able to identify a constraint violation and resolve it in a way that’s consistent with the true dynamics.

---

> ### Author Response · Authors · 2021-11-17
>
> **The novelty is weak. At its core, this paper applies the pipeline in (Sanchez-Gonzalez et al., 2020) … The iterative gradient-based solver has been proposed by Yang et al. (2020) in terms of the iterative projections along the gradient direction of the constraint network. ... Augmenting the input of the constraint function with Y ... seems minor and a straightforward enhancement.**
>
> We strongly disagree that the novelty is weak:
> - W.r.t to Sanchez-Gonzalez et al, we indeed follow the Processor-Updater pipeline, where we completely replace the Processor with our constraint model. The Updater is effectively only adding the predicted velocities to positions. The Processor-Updater pipeline is a simple idea that previous works did not claim as a contribution. We use a similar graph network as in Sanchez-Gonzalez et al. with different inputs, outputs and wrapped inside a constraint solver, that results in completely different inductive biases (see Figure 1’s (b) (Sanchez-Gonzalez et al.) versus (c) and (d) (ours)).
>
> - Wrt to Yang et al (2020) while we drew inspiration from that work, there are a number of innovations in our method:
>     - Using a GNN to implement a constraint function instead of an MLP (optionally with a hard-code grouping scheme). Using a GNN is not trivial because it introduces a number of questions about how to define a scalar function of a graph, which Yang et al didn’t have to address. The MLP model by Yang et al. is not feasible to run on the systems with >1000 nodes like BoxBath.
>
>     - Using a constraint function which takes not only the proposed future state that is being optimized as input, but also the most recent past state(s) as additional context. This allows our approach to model the dynamics as a constraint. Yang et al. relies on an Euler step (including hardcoding acceleration due to gravity) to move the dynamics forward in time to calculate the initial proposed future state and then evaluates the constraint only on the proposed future state being optimized, without context about the past states. This means Yang et al. model applicability is limited to a narrow set of physical simulations, for which it is possible to infer how to best correct an error introduced by the Euler step without context information . Consider this diagram for failure modes of Yang et al. [png file](https://drive.google.com/file/d/1JwU9cH0ap2hwnjfFzz6QzWTo8zV_Yt5R/view?usp=sharing). If a ball passes through a wall in Yang et al’s Euler step, the constraint cannot correct it because it is not aware of where the ball came from. By contrast, our method can prevent such incorrect dynamics because the constraints depend on both past and future states. Whether the modification itself is minor or straightforward doesn’t seem relevant: it’s a principled solution that makes a profound difference in performance between Yang et al. and our method.
>
>     - Our constraint function wasn’t defined as a zero point, or satisfied with root-finding. We defined the constraint as the minimum of a function and used gradient descent to satisfy it. We experimented with zero-point finding via the same Fast Projections algorithm used by Yang et. al. and found it much harder to train (see the variance of random seeds in figure 3, and trends in Fig B.4.) with deeper models in this more complex setting of modeling the time dynamics of a constraint. Materially, the use of minimization allowed us to manually add hand-designed constraints at test time.
>
>     - We demonstrated the unique advantages of the constraint model: (1) adding hand-designed constraints at inference time (2) improving generalization accuracy at inference time by using more solver steps. None of these experiments were shown in the literature, including Yang et al.
>
>     - We used a scheme for making the constraint function translation-invariant, but not providing as input absolute positions, but relative ones, as well as velocities. Yang et al’s model acts on the absolute positions and is not translation invariant.
>
> - We introduced another novel model, the Iterative GNN. Our intent was to try to interpolate between the explicit vs constraint-based simulators, to separately investigate the effects of the repeated iterations and using a learned constraint.
>
> Yang et al. is not necessarily expected to be applicable to these domains because of the reasons outlined above (it is not a fully learned simulator, but a learned projector) hence the performance is quite poor in our domains.We bridged the gap between our model and Yang et al (2020)’s, by proposing additional model hybrids similar to Yang et. al. (using Fast Projection, using MLPs instead of GNNs). We can include the results of the exact original Yang et al (2020) model on our environments in our paper, if reviewers would like.

---

### Official Review · Reviewer_kLRf · 2021-11-05

**Correctness:** 3
**Technical Novelty And Significance:** 2
**Empirical Novelty And Significance:** 2
**Recommendation:** 6
**Confidence:** 4

**Main Review:**

[Strength]

This paper tackles an important question of how we can incorporate prior knowledge in the form of explicit physical constraints in the learning-based simulators to enable better generalization.

The experiments on four environments show that the proposed method can deliver better prediction results than unconstrained baselines.


[Weakness]

While I like the direction this paper is going, I have concerns regarding the expressiveness of the experiments and the missing details of the method.

Prior methods on learning-based physics simulators have shown results on a set of much larger-scale and more complex environments involving fluids on rough terrain, fabrics with novel geometries, etc. [1, 2]. The experiment environments used in this paper may be a bit too simple compared to what's out there in the literature, making it hard to know how the method works in larger and more complicated scenarios.

Continuing from my previous point, [1] showed generalization to environments with drastically different geometry than seen during training, and [2] showed that the model could scale up to significantly larger and more complex cloth than seen during training. Adding constraints based on our understanding of physics is supposed to improve the model's generalization ability. As a result, I don't think the current experiments in the paper are enough to demonstrate the benefit of the constrained optimization process. The authors should consider including concrete experimental evidence on how the incorporated constraints may lead to even better/larger-scale generalization than what's already shown in the literature.

The authors should also consider including more details on how they construct the constraint function, f_C, e.g., for the fluid, rigid object, boundary conditions, etc. Without further details, it is hard for me to imagine how they are defined and implemented.

Related to my previous point, do we need any assumptions on f_C other than being differentiable? For example, for discontinuous events like contacts, I imagine we can differentiate through the LCP constraints, but how useful are the gradients?

How much more computing resources and time are needed to apply gradient descent on the constraint function? Multi-step message passing and solver iterations do not come for free. It may significantly increase forward prediction time for each time step. Therefore, it is essential to provide the time spent on each forward pass for different design choices and discuss the trade-off between the performance gain and the decrease in computational efficiency.

According to Figure 3, it is a bit hard to know which of the following three methods works better: (i) Iterative GNN, (ii) C-GNS-GD, and (iii) C-GNS-FP. When giving a new scenario, is there a way to know which one we should use, or should we try all of them and choose the one that works the best?

How did you choose the weight on the constraint term when incorporating novel constraints at test time? From the video, the rope is jittering. What might be the reason, and is it possible to resolve it? [1] shows generalization results of fluid simulation on unseen terrains much different from what the model was trained on. [3] also showed examples of generalization to unseen obstacle configurations. How would you compare the way your paper and [1, 3] incorporate new boundary constraints during testing?


[1] Benjamin Ummenhofer, Lukas Prantl, Nils Thuerey, Vladlen Koltun, "Lagrangian Fluid Simulation with Continuous Convolutions"
[2] Tobias Pfaff, Meire Fortunato, Alvaro Sanchez-Gonzalez, Peter W. Battaglia, "Learning Mesh-Based Simulation with Graph Networks"
[3] Alvaro Sanchez-Gonzalez, Jonathan Godwin, Tobias Pfaff, Rex Ying, Jure Leskovec, Peter W. Battaglia, "Learning to Simulate Complex Physics with Graph Networks"


===================

[Post Rebuttal]

I thank the authors for the detailed feedback, which addressed most of my concerns. I hope the authors can incorporate the response into the manuscript to improve its clarity. I'm happy to raise my score from 5 to 6.

**Summary Of The Paper:**

This paper aims to add explicit/human-defined constraints to learning-based simulation frameworks, where a learned constraint function implicitly regularizes the dynamics, and future predictions are generated via a constraint solver. The authors built the framework on top of graph neural networks (GNNs) to capture the compositionality of the underlying system and enforce the constraint using an implicit constraint function optimized via gradient descent.

The authors tested the proposed method in four physical simulation environments, including rope, bouncing balls, bouncing rigids, and BoxBath. Experiment results show that the proposed C-GNS has a competitive or better performance compared to prior learned simulators. In addition, they have also demonstrated that C-GNS can generalize to unseen, hand-designed constraints by applying more solver iterations than experienced during training to improve the accuracy on larger systems.

**Summary Of The Review:**

While I like the direction this paper is going, I have concerns regarding the expressiveness of the experiments compared to previous work on learning-based physics simulation, e.g., extreme extrapolation generalization on systems much larger than what the model was trained on.

There are also a lot of missing implementation details, making it hard to know (1) how the authors define and solve the constraints, e.g., for fluids, rigid objects, boundary conditions, etc., (2) how hard the constrained optimization problem really is, and (3) how does it affect the final performance.

As a result, I currently lean towards the rejection side.

---

> ### Author Response · Authors · 2021-11-17
> **Author's reply to the reviewer kLRf**
>
> **Prior methods … have shown results on a set of much larger-scale and more complex environments involving fluids on rough terrain, fabrics with novel geometries, etc. [1, 2]. The experiment environments used in this paper may be a bit too simple**
>
> We view the [1-3] papers as the culmination of a long line of work in learning explicit forward simulation (ie. Battaglia et al 2016, Chang et al. 2016, Sanchez et al 2018, Mrowca et al. 2018, Li et al. 2018;2019; etc, etc). Since they provide strong performance on key large-scale domains, we don’t see an immediate need to continue that line of work. Still, our results on smaller domains are clearly competitive with those methods, as are our results on the larger BoxBath environment, which [1, 3] tested as well.
>
> Rather, our primary intent here was to explore implicit, constrained-based learned simulators because they’re much less studied, and in principle they can offer other advantages over explicit forward approaches, e.g., controllable efficiency/accuracy trade-offs at test time by choosing the number of solver steps, generalizing to hand-designed constraints at test time which have not been observed during training, etc. Our results show clear advantages on these fronts. Since early learning simulation papers (eg. Battaglia et al. 2016 and Chang et al. 2016) scale well to larger, more complex environments in [1-3], and our method’s core architectural components are closely related, we anticipate that our present method will scale as well. However this is ultimately an empirical question, but because testing scalability rigorously is not trivial, we feel firmly that it’s well-justified to reserve such experiments on [1-3]’s large domains for future work.
>
> There are other reasons to use learned simulators in smaller domains, such as for object-centric reasoning and planning, from states or images/video–eg. Watters et al 2017, Janner et al 2018, Hamrick et al. 2018, Kipf et al 2019, Veerapaneni et al 2019, etc. So even if our approach were to not scale to engineering/graphics-level domains, there are many areas of ML that use learned simulators in the 5-50 object regime, which this could still benefit. In fact, Hamrick et al. 2018’s use of learned simulators with variable amounts of compute, decided by an internal RL policy, should directly benefit from our constraint-based approach’s ability to vary the number of solver steps at test time to strike efficiency/accuracy trade-offs.
>
> We will add text summarizing the above points in the introduction and Discussion.

---

> ### Author Response · Authors · 2021-11-17
> **Author's reply to the reviewer kLRf**
>
>
> **[1] showed generalization to environments with drastically different geometry than seen during training, and [2] showed that the model could scale up to significantly larger and more complex cloth than seen during training. … The authors should consider including concrete experimental evidence on how the incorporated constraints may lead to even better/larger-scale generalization than what's already shown in the literature.**
>
> There are two points to address here: (a) generalization to different geometry in [1], and (b) generalization to larger cloths in [2].
>
> For (a), [1] showed generalization to rigid collisions with different geometries, however their model was trained on rigid collisions. There has never been a report of [1] and similar models simulating rigid collisions if not trained on rigid collisions. This makes sense: the neural nets do not know how to interpret the “rigid” features that are provided as node attributes, let alone know how rigid collision dynamics should unfold in time. By contrast, in our hand-designed constraints experiments we had not trained the model on rigid collisions–they were only provided the rigid constraints at test time. Thus our model, which had never experienced an example of rigid dynamics, was nonetheless able to simulate it. In other words, its dynamics predictions generalized to the concept of rigid collision itself, given an appropriate constraint function. Again this makes sense: our model had learned the dynamics via its constraint function, and so incorporating new constraints at test time by adding them to the learned constraint function was compatible with the semantics imposed by the constraint-solving inductive bias. Generally, for explicit learned simulators like [1], it isn’t clear how to apply an explicit constraint objective without performing some kind of constrained optimization in the forward pass, which is effectively what we’re proposing.
>
> For (b), this is a very good point. In [2] and other papers they showed generalization to larger systems, so a natural question is: does our proposed approach offer any advantages? We partially addressed this in our previous response (ie. we tested BoxBath)--please see above for details. One thing to also note is that in [2] and other papers have typically shown videos or other demonstrations of the model generalizing to larger domains, but don’t report quantitative generalization results or compare to different approaches.
>
> That said, after reading this comment we realized we had decided not to include a comparison which might address this to some extent, and which we’re happy to add to the manuscript if the reviewer believes it’s helpful. We analyzed rollout generalization performance of our model compared to a close proxy to [2], with 10 message-passing layers on the longer (20-node) rope. A figure of this comparison, with the black line representing the [2] analog, is provided at this link: [png file](https://drive.google.com/file/d/1RVkqafklq2URDdems2DNVi736nGMTlx0/view?usp=sharing). We find that [2]’s MSE on this generalization experiment is roughly +50% larger than our C-GNS-GD’s. This is because ours can benefit from using additional solver iterations at test time, specifically using 7 or more optimization iterations at test time. This pattern of generalization has not been reported before, and we feel it represents an important advance. We can add this to the bottom-right plot of Figure 7a.

---

> ### Author Response · Authors · 2021-11-17
> **Author's reply to the reviewer kLRf**
>
> **The authors should also consider including more details on how they construct the constraint function, f_C, e.g., for the fluid, rigid object, boundary conditions, etc. Without further details, it is hard for me to imagine how they are defined and implemented.**
>
> This could be a misunderstanding, but just so we’re on the same page, our approach doesn’t require constructing any specific constraint functions (of course we did construct constraint functions in the later experiments about how novel hand-constructed constraints can be added at test time). The constraint function within our model (f_C) is a neural network that maps the previous and proposed state to a scalar value, which is minimized by gradient descent in the forward pass of the model. In other words, we impose an inductive bias that the output is that which minimizes some constraint function, and use the training process to determine what that constraint function represents. Thus, our approach does not require knowing anything about the actual constraints that govern the data, such as fluid, rigid or boundary conditions..
>
> &nbsp;
>
> **Do we need any assumptions on f_C other than being differentiable? For example, for discontinuous events like contacts, I imagine we can differentiate through the LCP constraints, but how useful are the gradients?**
>
> The only other assumptions about f_C (besides being differentiable, which neural networks obviously satisfy) is that it produces a scalar output and has a minimum, so it can be minimized by gradient descent. To ensure the constraint has a minimum, we squared the scalar output of f_C (although it actually turns out this wasn’t necessary, as we show in figure B.5 c-d).
>
> The point regarding contact interactions and LCP is interesting. We assume that in order for the model to learn a smooth constraint function which can be used to predict contact dynamics, the constraint function may adopt a strategy analogous to SLCP, where the discontinuity/non-differentiability is relaxed to a smooth approximation.
>
> &nbsp;
>
> **How much more computing resources and time are needed to apply gradient descent on the constraint function?**
>
> Yes, this is an important question. But again, the focus of this paper was on whether constrained-based simulation could compete with explicit forward methods, and exploring the unique advantages of our approaches over others. We provided comparisons with the Forward and Iterative GNN models (ie. Figure 4) which were intended to show how different numbers of message-passing steps and solver iterations traded off with performance, though we didn’t measure wall-clock time in the work.
>
> But our rough analysis (assuming no gradient checkpointing, and training through the solver) suggests that the running times (at inference and training), and the space requirements at training, are roughly:
> Forward Predictor: c * M
> Forward Iterative Predictor: c* M * N
> Constraint-based Predictor: c * M * N * b
> The M is the number of message passing steps, N is the number of solver iterations, c is a common constant for all three models, and b is between [1, 2). The b factor accounts roughly for the extra time it takes to compute the vector-Jacobian product of the neural networks with respect to the inputs (roughly 2x), but down-scaled because producing a scalar output from the constraint GNN means substituting a (smaller) weight vector for the (larger) final weight matrix in the other models.
>
> Note that while apparently the “Constraint-based Predictor” is slower by a factor of b * N compared to the “Forward Predictor”, on the training distribution the “Forward Predictor” typically needs a larger M to match the performance of the “Constraint-based Predictor”, as shown in Figure 4. This makes sense because the solver iterations should, in principle, be able to play a similar role as the message-passing steps. We will add this analysis and discussion to the Appendix.
>
> However, it’s also worth emphasizing that while compute efficiency is important, there are lots of directions here that should be explored in future work–eg. solvers with different convergence properties, adaptive numbers of solver steps (eg. Figure 7), etc.--which make it difficult to make a clear claim about efficiency without exhaustive exploration. Also, some applications, such as with small systems, may not be bottlenecked by the learned simulator’s compute, thus differences in the simulator’s compute may not make a material difference within larger architectures.. But again, this is an important question, and we tried to use Figure 4 to address it to a first approximation.

---

> ### Author Response · Authors · 2021-11-17
> **Author's reply to the reviewer kLRf**
>
> **According to Figure 3, it is a bit hard to know which of the following three methods works better: (i) Iterative GNN, (ii) C-GNS-GD, and (iii) C-GNS-FP. When giving a new scenario, is there a way to know which one we should use, or should we try all of them and choose the one that works the best?**
>
> Good question, we didn’t offer much prescription to readers here, which we’ll add as outlined below. As an aside, we also reported the numerical results (MSE and error bars over 5 seeds) for 1-step, 10-step and full rollout in Suppl. table B.1, which is the same data as in Figure 3.
>
> Our general conclusion is that (i) and (ii) offer best performance. For a new scenario where performance is prioritized, both should be tried.
> We concluded that (ii) offers unique advantages at test time: supporting adaptive computation, generalizing well to larger systems, and incorporating hand-designed constraints. So if these are desired, we suggest using (ii).
> We find that model (iii) is not competitive. It outperformed other methods in 1-step MSE on 1 task (by median performance), but everywhere else its performance was much worse. Moreover, this model was less stable to train: see high variance across the seeds in Figure 3 for most domains, including the BoxBath domain (where median performance was better). Moreover, Figure B.4 shows that the validation error degrades if trained with more iterations.
>
> Of course all three of these models were introduced by our work. Even model (iii) is distinct from Yang et al. 2020’s Neural Projections model, because we used a GNN rather than MLP, and we also conditioned the constraint function (f_C) on both the previous state and the proposed state, which they did not do.
>
> In our early exploration, we also experimented with the exact model from Yang et al 2020. However, its performance was far worse than for any of our models, and we saw little reason to report direct comparisons. Rather, we decided to report performance of a stronger variant, where we augmented the Yang et al. model with GNNs and conditioning on the past states (C-MLP-FP).
>
> &nbsp;
>
> **How did you choose the weight on the constraint term when incorporating novel constraints at test time? From the video, the rope is jittering. What might be the reason, and is it possible to resolve it?**
>
> Because the model is trained in a way that assumes a unique future state, the minimum of the constraint function is a point, rather than a high dimensional manifold. Note that in Figure 6 the model has *never observed* the collision between the rope and the obstacle at training time. When an extra constraint is added, the two constraints are conflicting, in other words, the minimum of the joint function does not overlap with the minimum of any of the individual constraints, causing the jittering.
>
> We treated the choice of weights as a hyperparameter and did a small sweep over values to determine which was best qualitatively. Because it’s only a scalar, and it allows tuning the extent to which the hand-crafted constraints are enforced in the simulation to suit a user’s interests, we don’t view this as a weakness.

---

> ### Author Response · Authors · 2021-11-17
> **Author's reply to the reviewer kLRf**
>
> **[1] shows generalization results of fluid simulation on unseen terrains much different from what the model was trained on. [3] also showed examples of generalization to unseen obstacle configurations. How would you compare the way your paper and [1, 3] incorporate new boundary constraints during testing?**
>
> As we noted above, in [1-3] the models were trained on the rigid collisions and tested on collisions against more complex geometries. As the interactions between wall and fluid particles are local, it is reasonable to expect the generalization to a new global configuration which locally looks like situations observed during training. We suspect that these models will not generalize to a different type of boundary constraints, such as if the walls are not represented as a line of dense particles.
>
> In contrast, in figure 6 our model had never experienced collisions between the rope and any object during training. Thus, we were able to introduce the concept of collision itself via an extra constraint function *at test time with no modifications to the model*.  If desired, we can use the same approach to add other constraints not observed during the training, such as collision with a soft object or preserving the length of the rope. In contrast, for forward models like [1,3] It is not possible to jointly optimize the dynamics and the satisfaction of the new constraints. Note that by making use of locally connected GNNs in our approach, our model benefits from both the types of generalization presented in [1-3], and this type of zero-shot generalization described in this paragraph.

---

### Author Response · Authors · 2021-11-23
**Adding Yang et al., 2020 baseline**

To address the reviewer's concerns about Yang et al. model, we have included the following models into the bar plots in Figure 3:

[Link to the plot (png file)](https://drive.google.com/file/d/1243w09ge45wb5FUGiXDclADEoe9FhQw-/view?usp=sharing)

- Yang et al (MLP) (light orange) -- the original model from Yang et al.
- Yang et al (GNN instead of MLP) (dark orange) -- same model with GNN projector, the baseline (1) suggested by reviewer AXAA.

The comparison of "C-MLP-FP" and "Yang et al (MLP)" models:
- The constraint for Yang et al (MLP) takes only the proposed future state Y
- The constraint in C-MLP-FP takes concatenated proposed state Y and future states X<=t
- Both optimize the absolute positions of the nodes as in Yang et al, 2020 paper
- In both, the constraint is parameterized by an MLP


The comparison of "C-GNS-FP" and "Yang et al (GNN instead of MLP)" models:
- The constraint for Yang et al (GNN instead of MLP) takes only the proposed future state Y
- The constraint in C-GNS-FP takes concatenated proposed state Y and future states X<=t
- Yang et al (GNN instead of MLP) optimize the absolute positions of the nodes as in Yang et al, 2020 paper
- C-GNS-FP do not observe the absolute positions, only the distances between each pair of nodes
- In both, the constraint is parameterized by a GNN

For the reasons that we explained to reviewers, Yang et al (MLP) model performs worse than C-MLP-FP; Yang et al (GNN instead of MLP) model performs 2 orders of magnitude worse than C-GNS-FP. This demonstrates that (1) augmenting the input with the past states X<=t has a significant impact on performance (2) the original model by Yang et al., 2020 (light orange) is not comparable to our weakest baselines presented in the paper.

We understand that we did not sufficiently explain our contributions wrt the previous paper. We are happy to include this baseline for rope, as well as the rest of the environments, if the reviewers still find it necessary for the motivation of this paper.

---

### Decision · Program_Chairs · 2022-01-20

**Decision:**

Reject

**Comment:**

The paper uses graph-based neural networks to ensure constraint-based simulation.  Even though the approach is a good one, it is only incremental w.r.t. the work published by Yang et al at NeurIPS in 2020; then, the experimental section is not convincing enough.

While the authors indicate their dissatisfaction with one of the reviewers' assessment, the overall reviews of the paper are not very positive.